# Fast John Ellipsoid Computation with Differential Privacy Optimization

Xiaoyu Li[1], Yingyu Liang[2,3], Zhenmei Shi[2], Zhao Song[4], Junwei Yu[5]
[1]University of New South Wales, [2]University of Wisconsin-Madison,
[3]The University of Hong Kong,
[4]The Simons Institute for the Theory of Computing at UC Berkeley,
[5]University of California, Berkeley
xiaoyu.li2@student.unsw.edu.au, yingyul@hku.hk, zhmeishi@cs.wisc.edu,
magic.linuxkde@gmail.com, yujunwei04@berkeley.edu

Determining the John ellipsoid - the largest volume ellipsoid contained within a convex polytope - is a fundamental problem with applications in machine learning, optimization, and data analytics. Recent work has developed fast algorithms for approximating the John ellipsoid using sketching and leverage score sampling techniques. However, these algorithms do not provide privacy guarantees for sensitive input data. In this paper, we present the first differentially private algorithm for fast John ellipsoid computation. Our method integrates noise perturbation with sketching and leverages score sampling to achieve both efficiency and privacy. We prove that (1) our algorithm provides $(\epsilon, \delta)$-differential privacy and the privacy guarantee holds for neighboring datasets that are $\epsilon_0$-close, allowing flexibility in the privacy definition; (2) our algorithm still converges to a $(1 + \xi)$-approximation of the optimal John ellipsoid in $\Theta(\xi^{-2}(\log(n/\delta_0) + (L\epsilon_0)^{-2}))$ iterations where $n$ is the number of data point, $L$ is the Lipschitz constant, $\delta_0$ is the failure probability, and $\epsilon_0$ is the closeness of neighboring input datasets. Our theoretical analysis demonstrates the algorithm's convergence and privacy properties, providing a robust approach for balancing utility and privacy in John ellipsoid computation. This is the first differentially private algorithm for fast John ellipsoid computation, opening avenues for future research in privacy-preserving optimization techniques.

## 1. Introduction

Determining the John ellipsoid (JE), involving calculating the best-fitting ellipsoid around a dataset, is a key challenge in machine learning, optimization, and data analytics. The John ellipsoid has been widely used in various applications, such as control and robotics [1, 2], obstacle collision detection [3], bandit learning [4, 5], Markov Chain Monte Carlo sampling [6], linear programming [7], portfolio optimization problems with transaction costs [8], and many so on. The objective of the John ellipsoid is to find the ellipsoid with the maximum volume that can be inscribed within a given convex, centrally symmetric polytope $P$, which is defined by a matrix $A \in \mathbb{R}^{n \times d}$ as follows.

**Definition 1.1** (Symmetric convex polytope, Definition 4.1 in [9]). *Let $A \in \mathbb{R}^{n \times d}$ be a matrix with full rank and $a_i^\top$ is the $i$-th row of $A$ for $i \in [n]$. The symmetric convex polytope $P$ is defined as*

$$P := \{x \in \mathbb{R}^d : |\langle a_i, x \rangle| \leq 1, \forall i \in [n]\}.$$

Recently, [10] introduced sketching techniques (Definition 3.10) to accelerate the John Ellipsoid computations, and [9] further speed up the John Ellipsoid algorithm by integrating the leverage score sampling method with the sketching technique, so they make John Ellipsoid computations can be run in practical time.

On the other hand, in many scenarios, it is essential and crucial to ensure that the ellipsoid's parameters are determined without revealing sensitive information about any individual data point while

Second Conference on Parsimony and Learning (CPAL 2025).

still allowing for the extraction of useful statistical information. For example, in bandit learning, we would like to provide privacy for each round of sensitive pay-off value while still getting some low-regret policy. Thus, in this work, we would like to ask and answer the following question,

*Can we preserve the privacy of individual data points in fast John Ellipsoid's computation?*

Our answer is positive by utilizing differential privacy (DP). By integrating differential privacy, our method provides a robust balance between data utility and privacy, enabling researchers and analysts to derive meaningful insights from data without compromising individual privacy. Moreover, the use of differential privacy in this context helps comply with data protection regulations, fostering trust in data-driven technologies.

## 1.1. Our Contributions

We first introduce the basic background of DP. Since for a polytope represented by $A \in \mathbb{R}^{n \times d}$, changing one row of the polytope matrix $A$ would result in a great variation in the geometric property. Therefore, using the general definition of neighborhood dataset fails to work. In our work, we define the $\epsilon_0$-closed neighborhood polytope, which ensures the privacy of Algorithm 1 with high accuracy. Thus, we consider two polytopes/datasets to be neighboring if they are $\epsilon_0$-close.

**Definition 1.2** (Neighboring polytopes). *Let $P, P'$ be two polytopes defined by $A, A'$, respectively. We say that $P$ and $P'$ are $\epsilon_0$-close if there exists exact one $i \in [n]$ such that $\|A_{i,*} - A'_{i,*}\|_2 \leq \epsilon_0$, and for all $j \in [n] \setminus \{i\}, A_{j,*} = A'_{j,*}$.*

Then, the differential privacy guarantee can be defined as the following.

**Definition 1.3** (Differential privacy). *A randomized mechanism $\mathcal{M} : \mathcal{D} \to \mathcal{R}$ with domain $\mathcal{D}$ and range $\mathcal{R}$ satisfies $(\epsilon, \delta)$-differential privacy if for any two neighboring dataset, $D, D' \in \mathcal{D}$ and for any subset of outputs $S \subseteq \mathcal{R}$ it holds that*

$$\Pr[\mathcal{M}(D) \in \mathcal{S}] \leq e^\epsilon \Pr[\mathcal{M}(D') \in \mathcal{S}] + \delta.$$

We propose the first algorithm for fast John ellipsoid computation that ensures differential privacy. Our work demonstrates the algorithm's convergence and privacy properties, providing a robust approach for balancing utility and privacy in John ellipsoid computation.

**Theorem 1.4** (Main Results, informal version of Theorem 4.1). *Let $\xi$ be the accuracy parameter, $\delta_0$ be the probability of failure, $L$ be the Lipschitz constant, and $n$ be the number of data points. Consider $\epsilon_0$-close neighboring polytopes. For all $\xi, \delta_0 \in (0, 0.1)$, when $T = \Theta(\xi^{-2}(\log(n/\delta_0) + (L\epsilon_0)^{-2}))$, we have that Algorithm 1 provides $(1 + \xi)$-approximation to John Ellipsoid with probability $1 - \delta_0$. Furthermore, for any $\epsilon \leq O(TL^2\epsilon_0^2)$, Algorithm 1 is $(\epsilon, \delta)$-differentially private for any $\delta > 0$ if we choose proper noise distribution. The running time of Algorithm 1 achieves $O((\mathrm{nnz}(A) + d^\omega)T)$, where $\omega \approx 2.37$ denotes the matrix-multiplication exponent [11–15].*

Our contributions can be summarized as the following:

- **DP optimization mechanism:** We provide a novel general DP-optimization analysis framework (Lemma 6.6) for truncated Gaussian noise, where we can show our final DP guarantee (Theorem 6.7) easily.
- **Fast DP-JE convergence:** We provide a convergence analysis (Theorem 7.2) for our fast DP-JE algorithm (Algorithm 1) under truncated Gaussian noise perturbation with DP guarantee (Theorem 4.1).
- **Perturbation analysis of the weighted leverage score:** We provide a comprehensive analysis of weighted leverage score perturbation (Lemma 5.1), which can be applied to many other fundamental problems of machine / statistical learning, e.g., kernel regression.

**Roadmap.** This paper is organized as follows: in Section 2, we study the related work about differential privacy, John Ellipsoid, leverage score, and sketching techniques. In Section 3, we define

notations used throughout our work. In Section 4, we demonstrate our main theorem about convergence and privacy of Algorithm 1. Then, in Section 5, we analyzed the Lipschitz of neighborhood polytope. Next, we demonstrate the differential privacy guarantee in our John Ellipsoid algorithm in Section 6. In Section 7, we show that our Algorithm 1 could solve John Ellipsoid with high accuracy and excellent running time. Finally, we conclude our work in Section 8.

## 2. Related Work

**John Ellipsoid Algorithm and Its Applications.** The John Ellipsoid Algorithm, initially proposed by [16], provides a powerful method for approximating any convex polytope by its maximum volume inscribed ellipsoid. This foundational work has spurred extensive research into optimization techniques for solving the John Ellipsoid problem within polynomial time constraints. Among the seminal contributions, [17, 18] introduced first-order methods, which significantly improved computational efficiency. Furthermore, [19–21] developed approaches utilizing interior point methods to enhance the precision and speed of solving the John Ellipsoid problem. Recent advancements have continued to push the boundaries of this algorithm. [10] employed fixed point iteration techniques, leading to the derivation of a more robust solution to the John Ellipsoid. Moreover, they introduced innovative sketching techniques that accelerated computational processes. Building on this, [9] integrated leverage score sampling into these sketching techniques, further optimizing the algorithm's performance, and [22] used quantum techniques to further speed up the computation of John Ellipsoids. The implications of the John Ellipsoid Algorithm extend far beyond theoretical mathematics, impacting various fields. In the realm of linear bandit problems, research by [4, 5] has shown significant advancements. Experimental design methods have also seen improvements due to contributions from [23, 24]. In linear programming, the algorithm has provided enhanced solutions, with notable work by [25]. Control theory applications have been advanced through research by [2], and cutting plane methods have been refined as demonstrated by [26]. The algorithm's influence in statistics is also noteworthy; for instance, it plays a critical role in Markov chain techniques for sampling convex bodies, as explored by [27] and developed for random walk sampling by [6, 28].

**Differential Privacy Analysis and Applications.** Differential privacy has become one of the most essential standards for data security and privacy protection since it was proposed in [29]. There are plenty of related work focusing on providing a guarantee for existing algorithms, data structures, and machine learning by satisfying the definition of differential privacy, such as [30–35, 35–63]. In addition, recently, there are emerging privacy mechanisms that improve traditional privacy guarantees, such as Gaussian, Exponential, and Laplacian mechanisms [64]. For example, [65] introduced a truncated Laplace mechanism, which has been demonstrated to achieve the tightest bounds among all $(\epsilon, \delta)$-DP distribution.

**Sketching and Leverage Score.** Our work improves the efficiency of the John Ellipsoid algorithm by leveraging sketching and score sampling. Sketching, a widely used technique, has advanced numerous domains, including neural network training, kernel methods [66, 67], and matrix sensing [68]. It has been applied to distributed problems [69, 70], low-rank approximation [71–73], and generative adversarial networks [74]. In addition, projected gradient descent [75], tensor-related problems [76, 77], and signal interpolation [78] have benefited significantly from sketching. Leverage scores, introduced by [79, 80], are pivotal in linear regression and randomized linear algebra, optimizing tasks such as matrix multiplication, CUR decompositions [81, 82], and tensor decompositions [82]. Moreover, leverage score sampling can be used in kernel learning [83]. Recent research has further extended the application of leverage score sampling. Studies by [66, 84–87] have demonstrated the ability to leverage score sampling to significantly enhance the efficiency of various algorithms and computational processes. These advancements underscore the versatility and effectiveness of leverage scores in optimizing performance across diverse fields.

# 3. Preliminary

Firstly, in Section 3.1, we introduce notations used in our work. Then, in Section 3.2, we demonstrate background knowledge about John Ellipsoid and the techniques we use to improve the running time of the John Ellipsoid algorithm, such as leverage score sampling and sketching. Finally, we introduce the techniques of leverage score sampling and sketching in Section 3.3.

## 3.1. Notations

In this paper, we utilize $\Pr[]$ to denote the probability. We use $\mathbb{E}[]$ to represent the expectation. For vectors $x \in \mathbb{R}^d$ and $y \in \mathbb{R}^d$, we denote their inner product as $\langle x, y \rangle$, i.e., $\langle x, y \rangle = \sum_{i=1}^{d} x_i y_i$. In addition, we denote $x_i^\top$ as the $i$-th row of $X$. We use $x_{i,j}$ to denote the $j$-th coordinate of $x_i \in \mathbb{R}^n$. We use $\|x\|_p$ to denote the $\ell_p$ norm of a vector $x \in \mathbb{R}^n$. For example, $\|x\|_1 := \sum_{i=1}^{n} |x_i|$, $\|x\|_2 := (\sum_{i=1}^{n} x_i^2)^{1/2}$, and $\|x\|_\infty := \max_{i \in [n]} |x_i|$. We use aux to represent auxiliary inputs in an adaptive mechanism. We use erf to denote the Gaussian error function.

For $n > d$, for any matrix $A \in \mathbb{R}^{n \times d}$, we denote the spectral norm of $A$ by $\|A\|$, i.e., $\|A\| := \sup_{x \in \mathbb{R}^d} \|Ax\|_2 / \|x\|_2$. We use $\|A\|_F$ to represent the Frobenius norm of $A$ , i.e., $\|A\|_F := (\sum_{i=1}^{n} \sum_{j=1}^{k} A_{i,j}^2)^{1/2}$. We use $\sigma_{\max}(A)$ to denote the maximum singular value of a matrix $A$ and use $\sigma_{\min}(A)$ to denote the minimum singular value of a matrix $A$. We use $\kappa(A) = \sigma_{\max}(A)/\sigma_{\min}(A)$ to denote the condition number of the matrix $A$. We use $\mathrm{nnz}(A)$ to denote the number of non-zero entries in matrix $A$.

## 3.2. Background Knowledge of John Ellipsoid

In this subsection, we introduced background knowledge about the John Ellipsoid algorithm, such as its definition, optimality criteria, and $(1 + \xi)$-approximate John Ellipsoid.

According to Definition 1.1, since $P$ is symmetric, the John Ellipsoid solution has to be centered at the origin. Any ellipsoid $E$ that is centered at the origin can be represented by the form $x^\top G^{-2} x \leq 1$, where $G$ is a positive definite matrix. Therefore, the optimal ellipsoid can be found by searching over the possible matrix $G$ as discussed in [10]:

**Definition 3.1** (Primal program of JE computation)**.** *The primal program of JE computation is*

$$\text{Maximize } \log((\det(G))^2),$$
$$\text{subject to} : G \succeq 0$$
$$\|Ga_i\|_2 \leq 1, \forall i \in [n].$$

[10] demonstrated that the optimal $G$ must satisfy the condition $G^{-2} = A^\top \mathrm{diag}(w)A$, where $A$ is a matrix and $w$ is a vector in $\mathbb{R}^n_{\geq 0}$. Consequently, by searching over all possible $w$, the dual optimization problem can be formulated:

**Definition 3.2** (Dual program of JE computation)**.** *The dual program of JE computation is*

$$\text{Minimize } \sum_{i=1}^{n} w_i - \log \det(\sum_{i=1}^{n} w_i a_i a_i^\top) - d, \tag{1}$$
$$\text{subject to} : w_i \geq 0, \ \forall i \in [n].$$

[88] shows that the optimal solution $w$ must satisfy the following conditions:

**Lemma 3.3** (Optimal solution, Proposition 2.5 in [88])**.** *Let $Q := \sum_{i=1}^{n} w_i a_i a_i^\top \in \mathbb{R}^{d \times d}$. A weight $w$ is optimal for program* (1) *if and only if*

$$\sum_{i=1}^{n} w_i = d,$$
$$a_j^\top Q^{-1} a_j = 1, \text{if } w_i \neq 0$$
$$a_j^\top Q^{-1} a_j < 1, \text{if } w_i = 0.$$

Other than deriving the exact John Ellipsoid solution, we give the definition of $(1+\xi)$-approximation to the exact solution, which is our goal in the fast DP-JE algorithm.

**Definition 3.4** $((1+\xi)$-approximate John Ellipsoid, Definition 4.3 in [9]). *For $\xi > 0$, we say $w \in \mathbb{R}^n_{\geq 0}$ is a $(1 + \xi)$-approximation of program (Eq. (1)) if $w$ satisfies*

$$\sum_{i=1}^n w_i = d, \text{ and } a_j^\top Q^{-1} a_j \leq 1 + \xi, \ \forall j \in [n].$$

**Lemma 3.5** $((1 + \xi)$-approximation is good rounding, Lemma 3.5 in [9]). *Let $P$ be defined as Definition 1.1. Let $w \in \mathbb{R}^n$ be a $(1 + \epsilon)$-approximation of program (Eq. (1)). Using that $w$, we define*

$$E := \{x \in \mathbb{R}^d : x^\top (\sum_{i=1}^n w_i a_i a_i^\top) x \leq 1\}.$$

*Then the following holds:*

$$\frac{1}{\sqrt{1+\epsilon}} \cdot E \subseteq P \subseteq \sqrt{d} \cdot E.$$

*Moreover, $\mathrm{vol}(\frac{1}{\sqrt{1+\epsilon}} E) \geq \exp(-d\epsilon/2) \cdot \mathrm{vol}(E^*)$ where $E^*$ is the exact John ellipsoid of $P$ and $\mathrm{vol}$ is volume function.*

**Remark 3.6.** *The exact John Ellipsoid solution, as defined by the optimality criteria in Lemma 3.3, provides a precise characterization of the ellipsoid. However, finding this exact solution can be computationally intensive due to its constraints. On the other hand, Definition 3.4 defines a relaxed version of the exact optimality condition, and Lemma 3.5 demonstrates that the approximate John Ellipsoid is a good approximation to the exact John Ellipsoid. Compared to finding the exact solution, the approximate solution only requires solving a less stringent optimization problem, which can significantly reduce computational complexity.*

## 3.3. Leverage Score and Sketching

In this subsection, we demonstrate the definition of leverage score, the convexity of the logarithm of the leverage score, and the sketching matrix, which are essential in our John Ellipsoid Algorithm 1 and convergence analysis.

**Definition 3.7** (Leverage score). *Given a matrix $A \in \mathbb{R}^{n \times d}$ with full column rank, we define its leverage score to be $A(A^\top A)^{-1} A^\top \in \mathbb{R}^{n \times n}$.*

The leverage scores measure the statistical importance of rows in a matrix. We also consider the weighted version of levearge scores, called Lewis weights.

**Definition 3.8** (Lewis weight). *The $\ell_p$ Lewis weights $w$ for matrix $A \in \mathbb{R}^{n \times d}$ is defined as the unique vector $w$ so that for all $i \in [n]$,*

$$w_i = w_i^{\frac{1}{2} - \frac{1}{p}} a_i^\top (A^\top \mathrm{diag}(w)^{1 - \frac{2}{p}} A)^{-1} a_i.$$

Given a matrix $A$, let $h : \mathbb{R}^n \to \mathbb{R}^n$ be the function defined as $h(w) = (h_1(w), h_2(w), \cdots, h_n(w))$ where $\forall i \in [n]$, we have

$$h_i(w) = a_i^\top (\sum_{j=1}^n w_j a_j a_j^\top)^{-1} a_i = a_i^\top (A^\top \mathrm{diag}(w) A)^{-1} a_i.$$

Hence, computing the $\ell_\infty$ Lewis weights is equivalent to solving the following fixed point problem:

$$w_i = w_i h_i(w), \ \forall i \in [n].$$

[10] observed that calculating $\ell_\infty$ Lewis weight is equivalent to determining the maximal volume inscribed ellipsoid in the polytope. By using the technique in [9], we find that $\ell_\infty$ Lewis weight is the weighted version of the standard leverage score. Therefore, by applying leverage score sampling techniques to Algorithm 1, we achieve speed up the calculation of Lewis weight in our fast John Ellipsoid algorithm similar to [9].

Now, we introduce the convexity lemma, used in demonstrating the correctness of Algorithm 1.

**Lemma 3.9** (Convexity, Lemma 3.4 in [10])**.** *For $i \in [n]$, let $\phi_i : \mathbb{R}^n \to \mathbb{R}$ be the function defined as*

$$\phi_i(v) = \log h_i(v) = \log(a_i^\top (\sum_{j=1}^n v_j a_j a_j^\top)^{-1} a_i).$$

*Then, $\phi_i$ is convex.*

Here, we give the main idea of the sketching matrix, which we utilized to speed up the running time to find John Ellipsoid in Algorithm 1.

**Definition 3.10** (Sketching)**.** *We define the sketching matrix $S_k \in \mathbb{R}^{s \times d}$ as a random matrix where each entry in the matrix is drawn i.i.d. from $\mathcal{N}(0, 1)$.*

# 4. Main Results

In this section, we demonstrate the main result of our work. By combining differential privacy and fast John Ellipsoid computation, we demonstrate that Algorithm 1 solved the John Ellipsoid problem with a differential privacy guarantee, high accuracy, and efficient running time (Theorem 4.1).

**Theorem 4.1** (Main Results, informal version of Theorem F.1)**.** *Let $v \in \mathbb{R}^n$ be the result of Algorithm 1. Define L as in Theorem 5.1. For all $\xi, \delta_0 \in (0, 0.1)$, when $T = \Theta(\xi^{-2}(\log(n/\delta_0) + (L\epsilon_0)^{-2}))$, the following holds for all $i \in [n]$:*

$$\Pr[h_i(v) \le (1 + \xi)] \ge 1 - \delta_0.$$

*In addition,*

$$\sum_{i=1}^n v_i = d.$$

*Thus, Algorithm 1 gives $(1 + \xi)$-approximation to the exact John ellipsoid.*

*Furthermore, suppose the input polytope in Algorithm 1 represented by $A \in \mathbb{R}^{n \times d}$ satisfies $\sigma_{\max}(A) \le \operatorname{poly}(n)$ and $\sigma_{\min}(A) \ge 1/\operatorname{poly}(n)$. Let $\epsilon_0 \le O(1/\operatorname{poly}(n))$ be the closeness of the neighboring polytopes defined in Definition 1.2 and $L \le O(\operatorname{poly}(n))$ be the Lipschitz defined in Theorem 5.1. Then for the $\epsilon_0$-close neighboring polytopes, for any $\epsilon \le O(TL^2\epsilon_0^2)$, Algorithm 1 is $(\epsilon, \delta)$-differentially private for any $\delta > 0$ if we choose the noise scale*

$$\sigma \ge \Omega(\frac{L\epsilon_0 \sqrt{T\log(1/\delta)}}{(1 - 2L\epsilon_0)\epsilon}).$$

*The runtime of Algorithm 1 is $O((\operatorname{nnz}(A) + d^\omega)T)$, where $\omega \approx 2.37$ represents the matrix multiplication exponent [11–15].*

Our Theorem 4.1 showed that our Algorithm 1 can approximate the ground-truth JE with a small error, i.e., $(1 + \xi)$, while our algorithm holds $(\epsilon, \delta)$-DP guarantee. Furthermore, our algorithm has the same running complexity, i.e., $\operatorname{nnz}(A) + d^\omega$ as the previous work [9].

Our algorithm uses three main techniques. First, in Line 14-15, we use the weighted leverage score sampling method to approximate the weighted matrix representation of the convex polytope, i.e., $B_k$ in Line 13. Then, in Line 16, we use a sketching matrix to reduce the dimension of the weighted matrix representation of the convex polytope. Finally, in Line 19-20, we inject our truncated Gaussian noise into the weighted leverage score to make our algorithm differential privacy.

# 5. Lipschitz Analysis of $\ell_\infty$-Lewis Weights

Before heading to the privacy analysis of Algorithm 1, we state the following theorem that derives the Lipschitz of the $\epsilon_0$-close neighborhood polytope. This analysis ensures that the variation in each iteration of converging to the final approximation of John Ellipsoid could be bounded by $L \cdot \epsilon_0$ in Algorithm 1. This bound is indispensable in the privacy analysis of Algorithm 1 in Section 6.

**Algorithm 1** Fast Algorithm for Differential Privacy Approximating John Ellipsoid (Fast DP-JE)

---

1: **procedure** FASTAPPROXDPJE($A \in \mathbb{R}^{n \times d}$, noise scale $\sigma$)
2:                                        ▷ A symmetric polytope given by $-\mathbf{1}_n \leq Ax \leq \mathbf{1}_n$, where $A \in \mathbb{R}^{n \times d}$
3:      $s \leftarrow \Theta(\xi^{-1})$
4:      $T \leftarrow \Theta(\xi^{-2}(\log(n/\delta_0) + (L\epsilon_0)^{-2}))$
5:      $\xi_0 \leftarrow \Theta(\xi)$
6:      $N \leftarrow \Theta(\xi_0^{-2} d \log(nd/\delta_1))$
7:      $\sigma \leftarrow \Theta(\frac{L\epsilon_0 \sqrt{T \log(1/\delta)}}{(1-2L\epsilon_0)\epsilon})$
8:      **for** $i = 1 \leftarrow n$ **do**
9:          Initialize $w_{1,i} = \frac{d}{n}$
10:      **end for**
11:      **for** $k = 1, \cdots, T-1$ **do**
12:          $W_k = \mathrm{diag}(w_k)$
13:          $B_k = \sqrt{W_k} A$
14:          Compute the $O(1)$-approximation for the leverage score of $B_k$
15:          Create a diagonal sampling matrix $D_k \in \mathbb{R}^{n \times n}$ based on leverage score
16:          Generate a random sketching $S_k \in \mathbb{R}^{s \times d}$ defined in Definition 3.10
17:          **for** $i = 1 \rightarrow n$ **do**
18:              $\overline{w}_{k+1,i} \leftarrow \frac{1}{s}\|S_k(B_k^\top D_k B_k)^{-1/2}\sqrt{w_{k,i}}a_i\|_2^2$
19:              Choose $z_{k+1,i} \sim \mathcal{N}^T(\mu, \sigma^2, [-0.5, 0.5])$
20:              $w_{k+1,i} = \overline{w}_{k+1,i}(1 + z_{k+1,i})$
21:          **end for**
22:      **end for**
23:      **for** $i = 1 \rightarrow n$ **do**
24:          $u_i = \frac{1}{T}\sum_{k=1}^T w_{k,i}$
25:      **end for**
26:      **for** $i = 1 \rightarrow n$ **do**
27:          $v_i = \frac{d}{\sum_{j=1}^n u_j} u_i$
28:      **end for**
29:      $V = \mathrm{diag}(v)$                            ▷ $V$ is a diagonal matrix formed from the elements of $v$
30:      **return** $V$ and $A^\top V A$         ▷ $(1+\xi)$-approximation of John Ellipsoid within the polytope
31: **end procedure**

---

**Theorem 5.1** (Lipschitz Bound for $\ell_\infty$-Lewis weights of $\epsilon_0$-close polytope, informal version of Theorem C.15). *Let $A, A' \in \mathbb{R}^{n \times d}$ where $a_i^\top$ and $a_i'^\top$ denote the $i$-th row of $A$ and $A'$, respectively, for $i \in [n]$ and suppose $A$ and $A'$ are only different in $j$-th row with $\|a_j - a_j'\|_2 \leq \epsilon_0$. Suppose that $W_k = \mathrm{diag}(w_k)$ where $\Omega(1) \leq w_{k,i} \leq 1$ for every $i \in [n]$. Let $f(w_k, A) := (f(w_k, A)_1, \ldots, f(w_k, A)_n)$ where $f(w_k, A)_i := w_i a_i^\top (A^\top W_k A)^{-1} a_i$ for every $i \in [n]$. Suppose that $\epsilon_0 \leq O(\sigma_{\min}(A))$. Then there exists $L = \mathrm{poly}(n, d, \kappa(A), \sigma_{\min}^{-1}(A), \sigma_{\max}(A))$ such that*

$$\|f(w_k, A) - f(w_k, A')\|_2 \leq L \cdot \epsilon_0.$$

*Proof sketch of Theorem 5.1.* The proof involves analyzing how small perturbations in the input matrix $A$ affect the resulting Lewis weights. By applying the perturbation theory of singular values and pseudo-inverses, we can bound the changes in Lewis weights caused by the small perturbation and demonstrate that the difference in the Lewis weights between the original and perturbed matrices is proportional to $\epsilon_0$, which ensures that the Lewis weights remain stable under small perturbations. $\quad\square$

## 6. Differential Privacy Analysis

In this section, we demonstrate our privacy analysis of Algorithm 1. Firstly, in Section 6.1, we introduce background knowledge on differential privacy about the sequential mechanism. Next, we show that Algorithm 1 achieves $(\epsilon, \delta)$-DP in Section 6.2.

## 6.1. Basic Definitions of Differential Privacy

In this subsection, we introduce the basic definitions of differential privacy and sequential mechanisms. Since our John Ellipsoid Algorithm 1 is an iterative algorithm that converges to the exact solution step by step, we need to use privacy techniques for the sequential mechanism [89], which means that the input of the algorithm depends on previous output. First, we listed privacy-related concepts about sequential mechanisms for the purpose of privacy analysis.

**Definition 6.1** (Sequential mechanism). *We define a sequential mechanism $\mathcal{M}$ consisting of a sequence of adaptive mechanisms $\mathcal{M}_1, \mathcal{M}_2, \cdots, \mathcal{M}_k$ where $M_i : \prod_{j=1}^{i-1} \mathcal{R}_j \times D \to \mathcal{R}_i$.*

Here, we give the definition of privacy loss, which measures the strength of privacy on a sequential mechanism.

**Definition 6.2** (Privacy loss). *For neighboring databases $D, D'$, a sequential mechanism $\mathcal{M}$, auxiliary input* aux, *and an outcome $o \in \mathcal{R}$, we define the privacy loss $c$ as the following*

$$c(o; \mathcal{M}, \mathsf{aux}, D, D') := \log \frac{\Pr[\mathcal{M}(\mathsf{aux}, D) = o]}{\Pr[\mathcal{M}(\mathsf{aux}, D') = o]}.$$

In our work, the privacy analysis of Algorithm 1 relies on bounding moments of loss of privacy in the sequential mechanism. Thus, we introduce $\alpha(\lambda)$ to denote the logarithm of moments of loss of privacy.

**Definition 6.3.** *We define the logarithm of moment generating function of $c(o; \mathcal{M}, \mathsf{aux}, D, D')$ as the following*

$$\alpha_{\mathcal{M}}(\lambda; \mathsf{aux}, D, D') := \log \mathop{\mathbb{E}}_{o \sim \mathcal{M}(\mathsf{aux}, D)} [\exp \lambda c(o; \mathcal{M}, \mathsf{aux}, D, D')].$$

**Definition 6.4.** *We define the maximum of $\alpha_{\mathcal{M}}(\lambda; \mathsf{aux}, D, D')$ taken over all auxiliary inputs and neighboring databases $D, D'$ as the following*

$$\alpha_{\mathcal{M}}(\lambda) := \max_{\mathsf{aux}, D, D'} \alpha_{\mathcal{M}}(\lambda; \mathsf{aux}, D, D').$$

Finally, we introduce truncated Gaussian noise we used to ensure the privacy of Algorithm 1. Unlike [89], which utilized standard Gaussian noise, we use truncated Gaussian instead. This is because truncated Gaussian noise could ensure that that error of Algorithm 1 caused by adding noise can be bounded by the accuracy parameter $\xi$, see details in Appendix E.4.

**Definition 6.5** (Truncated Gaussian). *We say that a random variable $z_n$ is from a truncated Gaussian distribution with mean 0 and variance $\sigma^2$ over the interval $[-0.5, 0.5]$, i.e., $z_n \sim \mathcal{N}^T(0, \sigma^2, [-0.5, 0.5])$, if its probability density function is defined as*

$$\mu_0(z_n) = \frac{1}{\sigma} \frac{\phi(\frac{z_n}{\sigma})}{\Phi(\frac{0.5}{\sigma}) - \Phi(\frac{-0.5}{\sigma})} \ \text{ for } \ z_n \in [-0.5, 0.5],$$

*where $\phi$ and $\Phi$ are pdf and cdf of the standard Gaussian.*

*Similarly, we use $\mu_1(z_n)$ to denote the pdf of $\mathcal{N}^T(\beta, \sigma^2, [-0.5, 0.5])$,*

$$\mu_1(z_n) = \frac{1}{\sigma} \frac{\phi(\frac{z_n - \beta}{\sigma})}{\Phi(\frac{0.5-\beta}{\sigma}) - \Phi(\frac{-0.5-\beta}{\sigma})} \text{ for } z_n \in [-0.5, 0.5].$$

*To simplify the pdf of truncated Gaussian, we define $C_\sigma, C_{\beta,\sigma}, \gamma_{\beta,\sigma}$ as $C_\sigma := \Phi(0.5/\sigma) - \Phi(-0.5/\sigma), C_{\beta,\sigma} := \Phi((0.5 - \beta)/\sigma) - \Phi((-0.5 - \beta)/\sigma)$, and $\gamma_{\beta,\sigma} := C_\sigma / C_{\beta,\sigma}$.*

## 6.2. Differential Privacy Optimization

Then, we can proceed to the privacy analysis of Algorithm 1. In Lemma 6.6, we show a general result on the upper bound about the moment of adding truncated Gaussian noise in a sequential mechanism.

**Lemma 6.6** (Bound of $\alpha(\lambda)$, informal version of Lemma D.6)**.** *Let $D, D' \in \mathcal{D}$ be the $\epsilon_0$-close neighborhood polytope in Definition 1.2. Suppose that $f : \mathcal{D} \to \mathbb{R}^n$ with $\|f(D) - f(D')\|_2 \leq \beta$. Let $z \in \mathbb{R}^n$ be a truncated Gaussian noise vector in Definition D.4. Let $\sigma = \min_i \sigma_i$ and $\sigma \geq \beta$. Then for any positive integer $\lambda \leq \gamma_{\beta,\sigma}/4$, there exists $C_0 > 0$ such that the mechanism $\mathcal{M}(d) = f(d) + z$ satisfies*

$$\alpha_\mathcal{M}(\lambda) \leq \frac{C_0 \lambda(\lambda+1)\beta^2 \gamma_{\beta,\sigma}^2}{\sigma^2} + O(\beta^3 \lambda^3 \gamma_{\beta,\sigma}^3 / \sigma^3).$$

*Proof Sketch of Lemma D.6.* Our goal is to bound the logarithm of the moment $\alpha_\mathcal{M}(\lambda)$. This is equivalent to the bound

$$\mathop{\mathbb{E}}_{z_n \sim \mu_1}[(\mu_1(z_n)/\mu_0(z_n))^\lambda] = \sum_{t=0}^{\lambda+1} \binom{\lambda+1}{t} \mathop{\mathbb{E}}_{z_n \sim \mu_0}[(\frac{\mu_1(z_n) - \mu_0(z_n)}{\mu_0(z_n)})^t]. \tag{2}$$

By basic algebra, the sum of the first two terms of Eq. (2) is 1. The third term can be bounded by

$$\frac{C_0 \lambda(\lambda+1)\beta^2 \gamma_{\beta,\sigma}^2}{\sigma^2}.$$

To bound the rest of the summation for $t \geq 4$, we consider 3 cases: $z_n \leq 0, 0 \leq z_n \leq \beta, z_n \leq \beta \leq 0.5$. We bound each term of Eq. (2) by separating them into the summation of the three cases. We derive that for the term $t = 4$, it could be bounded by

$$O(\beta^3 \lambda^3 \gamma_{\beta,\sigma}^3 / \sigma^3).$$

With our choice of $\lambda \leq 1/4\gamma_{\beta,\sigma}$ and $\sigma \geq \beta$, we find that all the higher order terms are dominated by the third term. Thus, we derive the upper bound for $\alpha_\mathcal{M}(\lambda)$.

$$\alpha_\mathcal{M}(\lambda) \leq \frac{C_0 \lambda(\lambda+1)\beta^2 \gamma_{\beta,\sigma}^2}{\sigma^2} + O(\beta^3 \lambda^3 \gamma_{\beta,\sigma}^3 / \sigma^3).$$

□

Therefore, combining Lemma 6.6 and the result of our Lipschitz analysis in Theorem 5.1, we demonstrate that Algorithm 1 achieved $(\epsilon, \delta)$-differential privacy.

**Theorem 6.7** (John Ellipsoid DP main theorem, informal version of Theorem D.7)**.** *Suppose that the input polytope in Algorithm 1 represented by $A \in \mathbb{R}^{n \times d}$ satisfies $\sigma_{\max}(A) \leq \operatorname{poly}(n)$ and $\sigma_{\min} \geq 1/\operatorname{poly}(d)$. There exists constants $c_1$ and $c_2$ so that given number of iterations $T$, for any $\epsilon \leq c_1 T \gamma \beta^2 \gamma_{L\epsilon_0,\sigma}$, Algorithm 1 is $(\epsilon, \delta)$-differentially private for any $\delta > 0$ if we choose*

$$\sigma \geq c_2 \frac{L\epsilon_0 \sqrt{T \log(1/\delta)}}{(1 - 2L\epsilon_0)\epsilon}$$

*Proof Sketch of Lemma D.7.* Since in Theorem 5.1, we demonstrate that $\epsilon_0$-close polytope can be bounded by $L\epsilon_0$. Thus, we substitute $\beta$ in Lemma 6.6 with $L\epsilon_0$. Since Algorithm 1 has $T$ iteration, we could apply the composition lemma for adaptive mechanism, described in Appendix D.4, to Lemma 6.6. Therefore, we have

$$\alpha(\lambda) \leq TC_0 L^2 \epsilon_0^2 \lambda^2 \gamma_{L\epsilon_0,\sigma}^2 \sigma^{-2}.$$

To satisfy the tail bound in Appendix D.4 and Lemma 6.6, we need to satisfy

$$TC_0 L^2 \epsilon_0^2 \lambda^2 \gamma_{L\epsilon_0,\sigma}^2 \sigma^{-2} \leq \lambda\epsilon/2,$$
$$\exp(-\lambda\epsilon/2) \leq \delta.$$

Therefore, by solving the system of inequality, we can show that Algorithm 1 is $(\epsilon, \delta)$-DP with our choice of $\sigma$ □

**Remark 6.8.** *[89] achieved differential privacy on stochastic gradient descent with privacy loss defined on sequential mechanism. To control the gradient perturbation, our moment bound $\alpha_\mathcal{M}(\lambda)$ uses the difference between the output of neighborhood datasets, while the moment bound in [89] needs the gradient to be less than or equal to 1. In addition, [89] only works when the algorithm adds noise for a small portion of datasets. In our setting, we derive the moment bound by adding noise for each data.*

# 7. John Ellipsoid Algorithm Convergence

In this section, we demonstrate that our Algorithm 1 could converge to a good approximation of exact John Ellipsoid with high probability under efficient running time.

First, we define $\widehat{w}_{k+1,i} := \|(B_k^\top B_k)^{-1/2}\sqrt{w_{k,i}}a_i\|_2^2$ and $\widetilde{w}_{k+1,i} := \|(B_k^\top D_k B_k)^{-1/2}\sqrt{w_{k,i}}a_i\|_2^2$. Intuitively, $\widehat{w}_{k+1,i}$ represents the ideal Lewis weight, and $\widetilde{w}_{k+1,i}$ denotes the Lewis weight computed using the leverage score sampling matrix. Then we introduce the technique of telescoping, which we utilize to separate the total relative error into ideal error $\widehat{w}_{k+1,i}/\widehat{w}_{k,i}$, leverage score sampling error $\widehat{w}_{k,i}/\widetilde{w}_{k,i}$, sketching error $\widetilde{w}_{k,i}/\overline{w}_{k,i}$, and error from truncated Gaussian noise $\overline{w}_{k,i}/w_{k,i}$.

**Lemma 7.1** (Telescoping lemma, informal version of Lemma E.3)**.** *Let $T$ denote the number of main iterations executed in our fast JE algorithm. Let $u$ be the vector generated during Algorithm 1. Then for each $i \in [n]$, we have*

$$\phi_i(u) \leq \frac{1}{T}\log\frac{n}{d} + \frac{1}{T}\sum_{k=1}^{T}\log\frac{\widehat{w}_{k,i}}{\widetilde{w}_{k,i}} + \frac{1}{T}\sum_{k=1}^{T}\log\frac{\widetilde{w}_{k,i}}{\overline{w}_{k,i}} + \frac{1}{T}\sum_{k=1}^{T}\log\frac{\overline{w}_{k,i}}{w_{k,i}}.$$

Then, we proceed to our convergence main theorem, which demonstrates that Algorithm 1 could converge a good approximation of John Ellipsoid with high accuracy.

**Theorem 7.2** (Convergence main theorem, informal version of Theorem E.12)**.** *Let $u \in \mathbb{R}^n$ be the non-normalized output of Algorithm 1, $\delta_0$ be the failure probability. For all $\xi \in (0,1)$, when $T = \Theta(\xi^{-2}(\log(n/\delta_0) + (L\epsilon_0)^{-2}))$, it holds that for all $i \in [n]$,*

$$\Pr[h_i(u) \leq (1+\xi)] \geq 1 - \delta_0.$$

*Moreover, it holds that*

$$\sum_{i=1}^{n} v_i = d.$$

*Therefore, Algorithm 1 finds $(1+\xi)$-approximation to the exact John ellipsoid solution.*

*Proof Sketch of Theorem 7.2.* Our goal is to show the leverage score $h_i(u)$ is less than or equal to $(1+\xi)$ with probability $1 - \delta_0$. Therefore, we need to bound each term in Lemma 7.1 by $\xi$. With the use of concentration inequality, we derive the high bound probability bound for the error of sketching, leverage score sampling, and truncated Gaussian noise. Combining the bound for each term together, we observe that when

$$T = \Theta(\xi^{-2}\log(n/\delta_0)),$$

we have

$$\log h_i(u) \leq \log(1+\xi), \forall i \in [n].$$

Next, by our choice of $v_i$ in Line 27 of Algorithm 1, we use the definition of leverage score $h_i(v)$ to show that

$$h_i(v) \leq (1+\epsilon).$$

$\square$

# 8. Conclusion

We presented the first differentially private algorithm for fast John ellipsoid computation, integrating noise perturbation with sketching and leverage score sampling. Our method provides $(\epsilon, \delta)$-differential privacy while converging to a $(1 + \xi)$-approximation in $O(\xi^{-2}(\log(n/\delta_0) + (L\epsilon_0)^{-2}))$ iterations. This work demonstrates a robust approach for balancing utility and privacy in geometric algorithms, opening avenues for future research in privacy-preserving optimization techniques.

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

# Appendix

**Roadmap.** In Section A, we introduce more related work in linear programming and privacy. Next, in Section B, we introduce background knowledge and tools involved in our work. Then, in Section C, we demonstrate the analysis about Lipschitz of $\epsilon_0$-close polytope. In Section D, we show that Algorithm 1 satisfies the DP guarantee. In Section E, we analyze the convergence and correctness of our John Ellipsoid algorithm. Finally, in Section F, we demonstrate our main theorem by combining privacy guarantee and correctness of Algorithm 1.

## A. More Related Work

In this section, we introduce more related work that inspires our research.

**Linear Programming and Semidefinite Programming**  Linear programming is a fundamental computer science and optimization topic. The Simplex algorithm, introduced in [90], is a pivotal method in linear programming, though it has an exponential runtime. The Ellipsoid method, which reduces runtime to polynomial time, is theoretically significant but often slower in practice compared to the Simplex method. The interior-point method, introduced in [91], is a major advancement, offering both polynomial runtime and strong practical performance on real-world problems. This method opened up a new avenue of research, leading to a series of developments aimed at speeding up the interior point method for solving a variety of classical optimization problems. John Ellipsoid has deep implication in the field of linear programming. For example, in interior point method, John Ellipsoid is utilized to find path to solutions [7]. The interior point method has a wide impact on linear programming as well as other complex tasks, such as [7, 92–104]. Moreover, the interior method and John ellipsoid are fundamental to solving semidefinite programming problems, such as [104–108].

Linear programming and semidefinite programming are widely applied in the field of machine learning theory, particularly in topics such as empirical risk minimization [99, 109, 110] and support vector machines [111, 112].

**Privacy and Security**  Data privacy and security have become a critical issue in the field of machine learning, particularly with the growing use of deep neural networks. As there is an increasing demand for training deep learning models on distributed and private datasets, privacy concerns have come to the forefront.

To address these concerns, various methods have been proposed for privacy-preserving deep learning. These methods often involve sharing model updates [113] or hidden-layer representations [114] rather than raw data. Despite these precautions, recent studies have shown that even if raw data remains private, sharing model updates or hidden-layer activations can still result in the leakage of sensitive information about the input, referred to as the victim. Such information leakage might reveal the victim's class, specific features [115], or even reconstruct the original data record [116–118]. This privacy leakage presents a significant threat to individuals whose private data have been utilized in training deep neural networks. Moreover, privacy and security have been studied in other fields in machine learning, such as attacks and defenses in federated learning [119–122], deep net pruning [123], language understanding tasks [124], alternating direction method of multipliers (ADMM) [125], and distributed learning [126].

## B. Baisc Tools

**Fact B.1** (Cauchy-Schwarz inequality). *For vectors $u, v \in \mathbb{R}^n$, we have*

$$\langle u, v \rangle \leq \|u\|_2 \cdot \|v\|_2$$

**Definition B.2** (Moment Generating Function of Gaussian). *Let $Z \sim \mathcal{N}(\mu, \sigma^2)$, the moment generating function of $Z$ is:*

$$M_Z(t) = \mathbb{E}[e^{tZ}] = \exp(t\mu + \frac{t^2\sigma^2}{2})$$

**Lemma B.3** (Hoeffding's bound, [127]). *Let $X_1, X_2, ..., X_n$ denote $n$ independent bounded variables in $[a_i, b_i]$. Let $X = \sum_{i=1}^{n} X_i$, then we have*

$$\Pr[|X - \mathbb{E}[X]| \geq t] \leq 2\exp(-2t^2/\sum_{i=1}^{n}(b_i - a_i)^2).$$

# C. Lipschitz Analysis on Neighborhood Polytopes

In this section, we delve into our analysis of the Lipschitz of $\epsilon_0$-close polytope. This analysis is crucial in showing the differential privacy guarantee of Algorithm 1. Firstly, in Section C.1, we provide some facts on matrix norm. Then, in Section C.2, we derive more matrix norm bounds on advanced matrix operation. Next, we demonstrate the bound on the norm of leverage score for neighborhood datasets in Section C.3.

## C.1. Basic Facts on Matrix Norm

In this section, we list basic facts about matrix norm.

**Fact C.1.** *Let $A \in \mathbb{R}^{n \times d}$ be a matrix. Then we have*

$$\|A\| \leq \|A\|_F.$$

**Fact C.2.** *Let $A \in \mathbb{R}^{n \times d}$ be a matrix where $a_i^\top$ is the $i$-th row of $A$. Then we have*

$$\|a_i\|_2 \leq \sigma_{\max}(A).$$

**Fact C.3.** *Let $A, B \in \mathbb{R}^{n \times d}, x \in \mathbb{R}^d$. Then the following two statements are equivalent:*

- $\|BB^\top - AA^\top\| \leq \epsilon$.

- $\|x^\top BB^\top x - x^\top AA^\top x\| \leq \epsilon \cdot x^\top x$.

**Lemma C.4** (Perturbation of singular value, [128]). *Let $A, B \in \mathbb{R}^{n \times d}$. Let $\sigma_i(A)$ denote the $i$-th singular value of $A$, then we have for any $i \in [d]$,*

$$\|\sigma_i(A) - \sigma_i(B)\| \leq \|A - B\|.$$

**Lemma C.5** (Perturbation of pseudoinverse, [129]). *Let $A, B \in \mathbb{R}^{n \times d}$. Then we have*

$$\|A^\dagger - B^\dagger\| \leq 2\max\{\|A^\dagger\|^2, \|B^\dagger\|^2\} \cdot \|A - B\|.$$

**Fact C.6.** *Let $A, B \in \mathbb{R}^{n \times d}, x \in \mathbb{R}^d$. Then we have*

- ***Part 1.*** $\|A\| = \|A^\top\| = \sigma_{\max}(A) \geq \sigma_{\min}(A)$.

- ***Part 2.*** $\|A^{-1}\| = \|A\|^{-1}$.

- ***Part 3.*** $\sigma_{\max}(B) - \|A - B\| \leq \sigma_{\max}(A) \leq \sigma_{\max}(B) + \|A - B\|$.

- ***Part 4.*** $\sigma_{\min}(B) - \|A - B\| \leq \sigma_{\min}(A) \leq \sigma_{\min}(B) + \|A - B\|$.

- ***Part 5.*** $\|Ax\|_2 \leq \|A\| \cdot \|x\|_2$.

## C.2. Bounds on Matrix Norm

In this subsection, given constraints on the spectral norm of the difference of matrices, we derive upper bounds on the spectral norm of operations defined by matrices.

Firstly, we show the constraints on the effect of perturbation on singular values.

**Lemma C.7.** *If the following conditions hold:*

- $\|A - B\| \leq \epsilon_0$.

- $\epsilon_0 \leq 0.1\sigma_{\min}(A)$.

*Then we have*

- $\sigma_{\max}(B) \in [0.9\sigma_{\max}(A), 1.1\sigma_{\max}(A)]$.

- $\sigma_{\min}(B) \in [0.9\sigma_{\min}(A), 1.1\sigma_{\min}(A)]$.

*Proof.* We can show

$$
\begin{aligned}
\sigma_{\max}(B) &\leq \sigma_{\max}(A) + \|A - B\| \\
&\leq \sigma_{\max}(A) + \epsilon_0 \\
&\leq \sigma_{\max}(A) + 0.1\sigma_{\max}(A) \\
&= 1.1\sigma_{\max}(A)
\end{aligned}
$$

where the first step follows from Fact C.6, the second and third steps follow from conditions, and the last step follows from basic algebra.

Next, we can show

$$
\begin{aligned}
\sigma_{\max}(B) &\geq \sigma_{\max}(A) - \|A - B\| \\
&\geq \sigma_{\max}(A) - \epsilon_0 \\
&\geq \sigma_{\max}(A) - 0.1\sigma_{\max}(A) \\
&= 0.9\sigma_{\max}(A)
\end{aligned}
$$

where the first step follows from Fact C.6, the second and third steps follow from conditions, and the last step follows from basic algebra.

Hence, we have

$$
\sigma_{\max}(B) \in [0.9\sigma_{\max}(A), 1.1\sigma_{\max}(A)].
$$

Similarly, we can show

$$
\sigma_{\min}(B) \in [0.9\sigma_{\min}(A), 1.1\sigma_{\min}(A)]
$$

using similar steps. $\square$

Then, we demonstrate the effect of singular value perturbation on gram matrix.

**Lemma C.8.** *If the following conditions hold:*

- $\|A - B\| \leq \epsilon_0$.

- $\epsilon_0 \leq 0.1\sigma_{\min}(A)$.

*Then we have*

$$
\|(B^\top B)^{-1}\| \in [0.7\sigma_{\max}(A)^{-2}, 1.3\sigma_{\min}(A)^{-2}]
$$

*Proof.* First, we show that

$$
\begin{aligned}
\|B^\top B\| &\leq \|B^\top\| \cdot \|B\| \\
&\leq \sigma_{\max}(B)^2 \\
&\leq 1.3\sigma_{\max}(A)^2.
\end{aligned}
$$

where the first step follows from basic algebra, the second step follows from Part 1. of Fact C.6, and the last step follows from Lemma C.7.

Hence, we have

$$\|(B^\top B)^{-1}\| = \|B^\top B\|^{-1}$$
$$\geq (1.3\sigma_{\max}(A)^2)^{-1}$$
$$\geq 0.7\sigma_{\max}(A)^2.$$

where the first step comes from Part 2. of Fact C.6, the second step follows from $\|B^\top B\| \leq 1.3\sigma_{\max}(A)^2$, and the last step follows from basic algebra.

Next, we can show

$$\|B^\top B\| \geq \|B^\top\| \cdot \|B\|$$
$$\geq \sigma_{\min}(B)^2$$
$$\geq 0.8\sigma_{\min}(A)^2.$$

where the first step comes from basic algebra, the second step comes from Part 1. of Fact C.6, and the last step follows from Lemma C.7.

Hence, we have

$$\|(B^\top B)^{-1}\| = \|B^\top B\|^{-1}$$
$$\leq (0.8\sigma_{\min}(A)^2)^{-1}$$
$$\leq 1.3\sigma_{\min}(A)^2.$$

where the first step derives from Part 2. of Fact C.6, the second step comes from $\|B^\top B\| \geq 0.8\sigma_{\min}(A)^2$, and the last step comes from basic algebra. $\square$

Next, we demonstrate the effect of perturbations on the spectral norm of the difference between matrices.

**Lemma C.9.** *If the following conditions hold*

- $\|A - B\| \leq \epsilon_0$

- $\epsilon_0 \leq 0.1\sigma_{\min}(A)$

*Then we have*

- $\|A^\top A - A^\top B\| \leq \sigma_{\max}(A) \cdot \epsilon_0$

- $\|A^\top B - B^\top B\| \leq 1.1\sigma_{\max}(A)\epsilon_0$

*Proof.* We can show that

$$\|A^\top A - A^\top B\| \leq \|A^\top\| \cdot \|A - B\|$$
$$= \sigma_{\max}(A) \cdot \|A - B\|$$
$$\leq \sigma_{\max}(A) \cdot \epsilon_0$$

where the first step follows from simple algebra, the second step follows from Fact C.6, and the last step follows from the assumption in the Lemma statement.

We can show that

$$\|A^\top B - B^\top B\| \leq \|B\| \cdot \|A^\top - B^\top\|$$
$$\leq 1.1\sigma_{\max}(A) \cdot \|A - B\|$$
$$\leq 1.1\sigma_{\max}(A) \cdot \epsilon_0$$

where the first step uses simple algebra, the second step utilizes Fact C.6, and the last step derives from the assumption in the Lemma statement. $\square$

Following the above lemma, we proceed to demonstrate the difference between two-gram matrices caused by perturbation on the singular value.

**Lemma C.10.** *If the following conditions hold*

- $\|A - B\| \leq \epsilon_0$

- $\epsilon_0 \leq 0.1\sigma_{\min}(A)$

*Then, we have*

$$\|A^\top A - B^\top B\| \leq 2.1\sigma_{\max}(A) \cdot \epsilon_0$$

*Proof.* We can show that

$$
\begin{aligned}
\|A^\top A - B^\top B\| &= \|A^\top A - A^\top B + A^\top B - B^\top B\| \\
&\leq \|A^\top A - A^\top B\| + \|A^\top B - B^\top B\| \\
&\leq \sigma_{\max}(A)\epsilon_0 + 1.1\sigma_{\max}(A)\epsilon_0 \\
&= 2.1\sigma_{\max}(A) \cdot \epsilon_0
\end{aligned}
$$

where the first step derives from simple algebra, the second step is by the triangle inequality, the last step derives from Lemma C.9, and the last step comes from basic algebra. $\square$

Then, we introduce condition numbers to bound the spectral norm on the inverse of the difference of two-gram matrices.

**Lemma C.11.** *If the following conditions hold*

- $\|A - B\| \leq \epsilon_0$.

- $\epsilon_0 \leq 0.1\sigma_{\min}(A)$.

*Then we have*

$$\|(AA^\top)^{-1} - (BB^\top)^{-1}\| \leq 8\kappa(A)\sigma_{\min}^{-3}(A)\epsilon_0.$$

*Proof.* We can show

$$
\begin{aligned}
\|(A^\top A)^{-1} - (B^\top B)^{-1}\| &\leq 2\max\{\|(A^\top A)^{-1}\|^2, \|(B^\top B)^{-1}\|^2\} \cdot \|(A^\top A) - (B^\top B)\| \\
&\leq 2 \cdot (1.3/\sigma_{\min}(A)^2)^2 \cdot \|(A^\top A) - (B^\top B)\| \\
&\leq 2 \cdot (1.3/\sigma_{\min}(A)^2)^2 \cdot (2.1\sigma_{\max}(A) \cdot \epsilon_0) \\
&\leq 8\kappa(A)\sigma_{\min}(A)^{-3}\epsilon_0,
\end{aligned}
$$

where the first step is by Lemma C.5, the second step is by Lemma C.7, the third step comes from Lemma C.10, the last step derives from $\kappa(A) = \frac{\sigma_{\max}(A)}{\sigma_{\min}(A)}$ and $2.1 \cdot (1.3)^2 \cdot 2.1 \leq 8$. $\square$

## C.3. Bounds of Lewis Weights on Neighborhood Datasets

In this subsection, we discuss bounds on Lewis weight and finally derive the Lipschitz bound for $\ell_\infty$ Lewis weights of $\epsilon_0$-close polytope.

Firstly, we derive the effect of the perturbation on one row of matrix $A$ on the spectral norm of the difference between weighted matrices.

**Lemma C.12.** *If the following conditions hold*

- *Let $A, A' \in \mathbb{R}^{n \times d}$.*

- *Let $a_i^\top$ denote the $i$-th row of $A$ for $i \in [n]$.*

- *Suppose $A$ and $A'$ is only different in $j$-th row, and $\|a_j - a'_j\|_2 \le \epsilon_0$.*
- *Suppose that $W_k = \mathrm{diag}(w_k)$ where $w_{k,i} \in [\Omega(1), 1]$ for every $i \in [n]$.*

*Then we have*

$$\|W_k^{1/2} A - W_k^{1/2} A'\| \le \epsilon_0.$$

*Proof.* Let $B = W_k^{1/2} A$ and $B' = W_k^{1/2} A'$. We have

$$
\begin{aligned}
\|B - B'\| = \|W_k^{1/2} A - W_k^{1/2} A'\| \\
\le \|W_k^{1/2} A - W_k^{1/2} A'\|_F \\
\le \sqrt{\sum_{i=1}^n \|w_{k,i} a_i - w_{k,i} a'_i\|_2^2} \\
= \|w_{k,i} a_j - w_{k,i} a'_j\|_2 \\
\le |w_{k,i}| \|a_j - a'_j\|_2 \\
= \epsilon_0
\end{aligned}
$$

where the first step comes from the definition of $B, B'$, the second step is the result of Fact C.1, the third step comes from the definition of Frobenius norm, the fourth step utilizes that $A$ and $A'$ only differs in $j$-th row, the fifth step derives from basic algebra, and the last step is from $w_{k,i} \in [0, 1]$ and $\|a_j - a'_j\|_2 \le \epsilon_0$. $\qquad\square$

Followed from Lemma C.12, we derive the bound for perturbation on $(A^\top W_K A)^{-1}$

**Lemma C.13.** *If the following conditions hold*

- *Let $A, A' \in \mathbb{R}^{n \times d}$.*
- *Let $a_i^\top$ denote the $i$-th row of $A$ for $i \in [n]$.*
- *Suppose $A$ and $A'$ is different in $j$-th row, and $\|a_j - a'_j\|_2 \le \epsilon_0$*
- *Suppose that $W_k = \mathrm{diag}(w_k)$ where $w_{k,i} \in [\Omega(1), 1]$ for every $i \in [n]$.*
- *Suppose that $\epsilon_0 \le O(\sigma_{\min}(A))$.*

*Then we have*

$$\|(A^\top W_k A)^{-1} - (A'^\top W_k A')^{-1}\| \le O(8\kappa(A) \sigma_{\min}^{-3}(A) \epsilon_0).$$

*Proof.* By Lemma C.12, we have

$$\|W_k^{1/2} A - W_k^{1/2} A'\| \le \epsilon_0.$$

We can show that

$$
\begin{aligned}
\|(A^\top W_k A)^{-1} - (A'^\top W_k A')^{-1}\| &\le 8\kappa(W_k^{1/2} A) \sigma_{\min}^{-3}(W_k^{1/2} A) \epsilon_0 \\
&\le O(8\kappa(A) \sigma_{\min}^{-3}(A) \epsilon_0).
\end{aligned}
$$

where the first step is the result of Lemma C.11 and the second step is from $w_{k,i} \in [\Omega(1), 1]$ for every $i \in [n]$. $\qquad\square$

Next, we introduce $f$, an algorithm that computes Lewis weight. And we derive the upper bound of change in $f$ caused by perturbation on input $A$.

**Lemma C.14.** *If the following conditions hold*

- *Let $A, A' \in \mathbb{R}^{n \times d}$.*

- *Let $a_i^\top$ denote the $i$-th row of $A$ for $i \in [n]$.*

- *Suppose $A$ and $A'$ is different in $j$-th row, and $\|a_j - a_j'\|_2 \leq \epsilon_0$*

- *Suppose that $W_k = \mathrm{diag}(w_k)$ where $w_{k,i} \in [\Omega(1), 1]$ for every $i \in [n]$.*

- *Let $f(w_k, A) := (f(w_k, A)_1, \ldots, f(w_k, A)_n)$*

- *Let $f(w_k, A)_i := w_i a_i^\top (A^\top W_k A)^{-1} a_i$ for $i \in [n]$.*

- *Suppose that $\epsilon_0 \leq O(\sigma_{\min}(A))$.*

- *Let $\epsilon_1 = \Theta(8\kappa(A)\sigma_{\min}^{-3}(A)\epsilon_0)$.*

*Then we have*

- ***Part 1.** For $i \neq j$, $|f(w_k, A)_i - f(w_k, A')_i| \leq \epsilon_1 \cdot \sigma_{\max}(A)^2$.*

- ***Part 2.** $|f(w_k, A)_j - f(w_k, A')_j| \leq \epsilon_1(\sigma_{\max}(A) + \epsilon_0)^2 + \epsilon_0 \sigma_{\min}(W_k^{1/2}A)^2(2\sigma_{\max}(A) + \epsilon_0)$*

*Proof.* **Proof of Part1.** For $i \neq j$, we have

$$
\begin{aligned}
|f(w_k, A)_i - f(w_k, A')_i| &= |w_{k,i} a_i^\top (A^\top W_k A)^{-1} a_i - w_{k,i} a_i^\top (A'^\top W_k A')^{-1} a_i| \\
&\leq |w_{k,i}| \cdot |a_i^\top (A^\top W_k A)^{-1} a_i - a_i^\top (A'^\top W_k A')^{-1} a_i| \\
&\leq |a_i^\top (A^\top W_k A)^{-1} a_i - a_i^\top (A'^\top W_k A')^{-1} a_i| \\
&\leq \epsilon_1 \cdot a_i^\top a_i \\
&= \epsilon_1 \cdot \|a_i\|^2 \\
&\leq \epsilon_1 \cdot \sigma_{\max}(A)^2
\end{aligned}
$$

where the first step follows from the definition of $f$, the second definition comes from basic algebra, the third step comes from $w_{k,i} \in [0, 1]$, the fourth step derives from Lemma C.13 and Fact C.3, the fifth step utilizes basic algebra, and the last step derives from Fact C.2.

**Proof of Part 2.** Next, we define

$$
\begin{aligned}
C_1 &:= a_j^\top (A^\top W_k A)^{-1} a_j - a_j'^\top (A^\top W_k A)^{-1} a_j', \\
C_2 &:= a_j'^\top (A^\top W_k A)^{-1} a_j' - a_j'^\top (A'^\top W_k A')^{-1} a_j'.
\end{aligned}
$$

We first bound $C_1$. We can show that

$$
\begin{aligned}
|C_1| &= |a_j^\top (A^\top W_k A)^{-1} a_j - a_j'^\top (A^\top W_k A)^{-1} a_j'| \\
&= |a_j^\top (A^\top W_k A)^{-1} a_j - a_j'^\top (A^\top W_k A)^{-1} a_j + a_j'^\top (A^\top W_k A)^{-1} a_j - a_j'^\top (A^\top W_k A)^{-1} a_j'| \\
&= |\underbrace{(a_j - a_j')^\top (A^\top W_k A)^{-1} a_i}_{C_3} + \underbrace{a_j'^\top (A^\top W_k A)^{-1} (a_i - a_j')}_{C_4}| \\
&\leq |C_3| + |C_4|.
\end{aligned}
$$

where the first step follows from the definition of $C_1$, the second and third steps follow from basic algebra, and the last step follows from the triangle inequality.

For $C_3$, we have

$$
\begin{aligned}
|C_3| &= |(a_j - a_j')^\top (A^\top W_k A)^{-1} a_i| \\
&\leq \|(a_j - a_j')\|_2 \cdot \|(A^\top W_k A)^{-1} a_i\| \\
&\leq \|(a_j - a_j')\|_2 \cdot \|(A^\top W_k A)^{-1}\| \cdot \|a_i\|_2
\end{aligned}
$$

$$\leq \epsilon_0 \cdot \sigma_{\min}(W_k^{1/2}A)^2 \cdot \sigma_{\max}(A)$$

where the first step comes from definition of $C_3$, the second step utilizes Cauchy-Schwarz inequality, the third step derives from Part 5 of Fact C.6, and the last step comes from Lemma assumptions, Fact C.2 and Part 2 of Fact C.6.

For $C_4$, we have

$$
\begin{aligned}
|C_4| &= |a_j'^\top (A^\top W_k A)^{-1} (a_j - a_j')| \\
&\leq \|a_j'\|_2 \cdot \|(A^\top W_k A)^{-1}(a_j - a_j')\| \\
&\leq \|a_j'\|_2 \cdot \|(A^\top W_k A)^{-1}\| \cdot \|a_j - a_j'\|_2 \\
&\leq (\|a_j\|_2 + \epsilon_0) \cdot \sigma_{\min}(W_k^{1/2}A)^2 \cdot \epsilon_0 \\
&\leq (\sigma_{\max}(A) + \epsilon_0) \cdot \sigma_{\min}(W_k^{1/2}A)^2 \cdot \epsilon_0
\end{aligned}
$$

where the first step comes from the definition of $C_3$, the second step is from Cauchy-Schwarz inequality, the third step derives from Part 5 of Fact C.6, and the fourth step is from $\|a_j - a_j'\| \leq \epsilon_0$ and Part 2 of Fact C.6, and the last step comes from Fact C.2.

Combining the bounds of $|C_3|$ and $|C_4|$, we have

$$|C_1| \leq \epsilon_0 \cdot \sigma_{\min}(W_k^{1/2}A)^2 \cdot (2\sigma_{\max}(A) + \epsilon_0)$$

We next bound $C_2$. We can show that

$$
\begin{aligned}
|C_2| &= |a_j'^\top (A^\top W_k A)^{-1} a_j' - a_j'^\top (A'^\top W_k A')^{-1} a_j'| \\
&\leq \epsilon_1 a_j'^\top a_j' \\
&\leq \epsilon_1 \|a_j'\|^2 \\
&\leq \epsilon_1 (\|a_j\| + \epsilon_0)^2 \\
&\leq \epsilon_1 (\sigma_{\max}(A)^2 + \epsilon_0)^2
\end{aligned}
$$

where the first step follows from the definition of $C_2$, the second step follows from Lemma C.11 and Fact C.3, and the third step follows from basic algebra, and the last step follows from $\|a_j - a_j'\| \leq \epsilon_0$.

We can show that

$$
\begin{aligned}
|f(w_k, A)_j - f(w_k, A')_j| &= |w_{k,i} a_i^\top (A^\top W_k A)^{-1} a_i - w_{k,i} a_j'^\top (A'^\top W_k A')^{-1} a_j'| \\
&= |w_{k,i} C_1 + w_{k,i} C_2| \\
&= |w_{k,i}||C_1 + C_2| \\
&\leq |C_1| + |C_2| \\
&\leq \epsilon_1 (\sigma_{\max}(A)^2 + \epsilon_0)^2 + \epsilon_0 \sigma_{\min}(W_k^{1/2}A)^2 (2\sigma_{\max}(A) + \epsilon_0)
\end{aligned}
$$

where the first step stems from the definition of $f$, the second step comes from the definition of $C_1, C_2$, the third step is from basic algebra, the fourth step comes from $w_{k,i} \leq 1$ and triangle inequality, and the last step derives from the bounds of $|C_1|$ and $|C_2|$. $\qquad \square$

Finally, we could bound the max difference between outputs for $f$ for two $\epsilon_0$-close input polytopes $A, A'$.

**Theorem C.15** (Lipschitz Bound for $\ell_\infty$ Lewis weights of $\epsilon_0$-close polytope, formal version of Theorem 5.1). *If the following conditions hold*

- *Let $A, A' \in \mathbb{R}^{n \times d}$ where $a_i^\top$ and $a_i'^\top$ denote the $i$-th row of $A$ and $A'$, respectively, for $i \in [n]$.*

- *Suppose that $A$ and $A'$ are only different in $j$-th row, and $\|a_j - a_j'\|_2 \leq \epsilon_0$*

- *Suppose that $W_k = \mathrm{diag}(w_k)$ where $w_{k,i} \in [0, 1]$ for every $i \in [n]$.*

- *Let $f(w_k, A) := (f(w_k, A)_1, \ldots, f(w_k, A)_n)$*
- *Let $f(w_k, A)_i := w_i a_i^\top (A^\top W_k A)^{-1} a_i$ for $i \in [n]$.*
- *Suppose that $\epsilon_0 \leq O(\sigma_{\min}(A))$.*
- *Let $L = \mathrm{poly}(n, d, \kappa(A), \sigma_{\min}^{-1}(A), \sigma_{\max}(A))$*

*Then, we can show*

$$\|f(w_k, A) - f(w_k, A')\|_2 \leq L \cdot \epsilon_0.$$

*Proof.* For proof purpose, we set $\epsilon_1 = 8(W_k^{1/2} A) \sigma_{\min}^{-3}(W_k^{1/2} A) \epsilon_0$. We can show that

$$
\begin{aligned}
\|f(w_k, A) - f(w_k, A')\|_2^2 &= \sum_{i=1}^n |f(w_k, A)_i - f(w_k, A')_i|^2, \\
&= \Big( \sum_{i \in [n] \setminus \{j\}} |f(w_k, A)_i - f(w_k, A')_i|^2 \Big) + |f(w_k, A)_j - f(w_k, A')_j|^2 \\
&\leq (n-1)(\epsilon_1 \cdot \sigma_{\max}(A)^2)^2 + (\epsilon_1(\sigma_{\max}(A) + \epsilon_0)^2 + \epsilon_0 \sigma_{\max}(A)^2 (2\sigma_{\max}(A) + \epsilon_0))^2 \\
&= L^2 \epsilon_0^2
\end{aligned}
$$

where the first step is from the definition of $\ell_2$ norm, the second step comes from basic algebra, the third step derives from Lemma C.14, and the last step comes from observing that the right-hand-side is the product of $\epsilon_0^2$ and a polynomial in other parameters. $\qquad \square$

# D. Differentially Private John Ellipsoid Algorithm

Firstly, in Section D.1, we present tools used in the proof of our main Lemma D.6 about bounding moments. Then, in Section D.2, we introduced truncated Gaussian noise and derived moments bound on truncated Gaussian about differential privacy. Next, in Section D.3, we proceed to our main theorem about the privacy of Algorithm 1. Finally, in Section D.4, we introduce the composition lemma used in our privacy proof.

## D.1. Facts and Tools

In this subsection, we demonstrate basic numerical and probability tools utilized in later proofs.

**Lemma D.1.** *Let $\mu_0$ be the probability density function of $N(0, 1)$, $z$ is a random variable with distribution $\mu_0$. For any $a \in \mathbb{R}$,*

$$\mathbb{E}_{z \sim \mu_0}[\exp(\frac{2az}{2\sigma^2})] = \exp(\frac{a^2}{2\sigma^2})$$

*Proof.* We proceed with the proof with the moment generating function of the Gaussian variable, described in Definition B.2.

Recall Definition B.2, we have

$$M_Z(t) = \mathbb{E}[e^{tZ}] = \exp(t\mu + \frac{t^2 \sigma^2}{2})$$

Therefore, we get

$$
\begin{aligned}
\mathbb{E}_{z \sim \mu_0}[\exp(\frac{2az}{2\sigma^2})] &= \mathbb{E}_{z \sim \mu_0}[\exp(\frac{az}{\sigma^2})] \\
&= \exp(0 + \frac{1}{2}(\frac{a}{\sigma^2})^2 \sigma^2) \\
&= \exp(\frac{a^2}{2\sigma^2})
\end{aligned}
$$

where the first step stems from simplification, the second step is by setting $\mu = 0$ and $t = \frac{a}{\sigma^2}$ in moment generating function of Gaussian variable, and the final step comes from basic algebra.

$\square$

**Fact D.2.** *For any $\sigma \geq 1$, we have*

$$\frac{1}{2}(\exp(1/\sigma^2) - 1) \leq \frac{1}{\sigma^2}$$

*Proof.* It's easy to show the inequality using the Taylor expansion of the exponential function. $\square$

## D.2. Moments Bound for Truncated Gaussian Noise

In this subsection, we first introduced the relevant definition of truncated Gaussian and defined some variables. Then, we proceed to the proof of Lemma D.6 about moment bound with truncated Gaussian noise.

Firstly, we introduce the truncated Gaussian we used to ensure the privacy of Algorithm 1.

**Definition D.3** (Truncated Gaussian). *Given a random variable $z_i$ with truncated Gaussian distribution $\mathcal{N}^T(\mu, \sigma^2, [-0.5, 0.5])$, its probability density function is defined as*

$$g(z_i) = \frac{1}{\sigma} \frac{\phi(\frac{z_i - \mu}{\sigma})}{\Phi(\frac{0.5 - \mu}{\sigma}) - \Phi(\frac{-0.5 - \mu}{\sigma})} \text{ for } z_i \in [-0.5, 0.5]$$

*where $\phi$ and $\Phi$ are pdf and cdf of the standard Gaussian.*

*We also define constants $C_\sigma, C_{\beta,\sigma}, k$ to simplify the pdf of $\mathcal{N}^T(0, \sigma^2, [-0.5, 0.5])$ and $\mathcal{N}^T(\beta, \sigma^2, [-0.5, 0.5])$*

$$C_\sigma := \Phi(0.5/\sigma) - \Phi(-0.5/\sigma)$$
$$C_{\beta,\sigma} := \Phi((0.5 - \beta)/\sigma) - \Phi((-0.5 - \beta)/\sigma)$$
$$\gamma_{\beta,\sigma} := \frac{C_\sigma}{C_{\beta,\sigma}}$$

Here, we give the definition of truncated Gaussian noise in vector form.

**Definition D.4** (Truncated Gaussian vector). *We define the truncated Gaussian noise vector $z = (z_1, z_2, \cdots, z_n)$, where each $z_i$ follows Definition 6.5*

Now, we'll derive the lower and upper bounds of $\gamma_{\beta,\sigma}$ for the purpose of differential privacy proof of Algorithm 1.

**Lemma D.5.** *Let $\gamma_{\beta,\sigma}$ be the one introduced in Definition 6.5. Given $\sigma \geq \beta$, the following bound of $\gamma_{\beta,\sigma}$ holds*

$$1 \leq \gamma_{\beta,\sigma} \leq \frac{1}{1 - 2\beta}$$

*Proof.* According to the definition of $\gamma_{\beta,\sigma}$, we have

$$\gamma_{\beta,\sigma} = \frac{\Phi(0.5/\sigma) - \Phi(-0.5/\sigma)}{\Phi((0.5 - \beta)/\sigma) - \Phi((-0.5 - \beta)/\sigma)}$$

Firstly, we consider the numerator,

$$\Phi(0.5/\sigma) - \Phi(-0.5/\sigma) = \frac{1}{2}(1 + \mathrm{erf}(\frac{0.5/\sigma}{\sqrt{2}})) - \frac{1}{2}(1 + \mathrm{erf}(-\frac{0.5/\sigma}{\sqrt{2}}))$$
$$= \frac{1}{2}(1 + \mathrm{erf}(\frac{0.5/\sigma}{\sqrt{2}})) - \frac{1}{2}(1 - \mathrm{erf}(\frac{0.5/\sigma}{\sqrt{2}}))$$

$$= \frac{1}{2}(\text{erf}(\frac{0.5/\sigma}{\sqrt{2}}) + \text{erf}(\frac{0.5/\sigma}{\sqrt{2}}))$$

$$= \text{erf}(\frac{0.5/\sigma}{\sqrt{2}})$$

where the first step comes from the definition of cdf for standard normal distribution, the second step comes from the property $\text{erf}(-x) = -\text{erf}(x)$, the third step utilizes basic algebra, and the last is by simplification.

We simplify the denominator similarly,

$$\Phi((0.5 - \beta)/\sigma) - \Phi((-0.5 - \beta)/\sigma) = \frac{1}{2}(1 + \text{erf}(\frac{(0.5 - \beta)/\sigma}{\sqrt{2}})) - \frac{1}{2}(1 + \text{erf}(\frac{(-0.5 - \beta)/\sigma}{\sqrt{2}}))$$

$$= \frac{1}{2}(1 - \text{erf}(\frac{(0.5 - \beta)/\sigma}{\sqrt{2}})) - \frac{1}{2}(1 - \text{erf}(\frac{(0.5 + \beta)/\sigma}{\sqrt{2}}))$$

$$= \frac{1}{2}(\text{erf}(\frac{(0.5 - \beta)/\sigma}{\sqrt{2}}) + \text{erf}(\frac{(0.5 + \beta)/\sigma}{\sqrt{2}}))$$

where the first step comes from the definition of cdf for standard normal distribution, the second step comes from the property $\text{erf}(-x) = -\text{erf}(x)$, and the third step utilizes basic algebra.

Combine them together, we have

$$\gamma_{\beta,\sigma} = \frac{2\text{erf}(\frac{0.5/\sigma}{\sqrt{2}})}{\text{erf}(\frac{(0.5-\beta)/\sigma}{\sqrt{2}}) + \text{erf}(\frac{(0.5+\beta)/\sigma}{\sqrt{2}})}$$

For the lower bound of $\gamma_{\beta,\sigma}$, we observe that as $\beta$ approaches to 0, $\gamma_{\beta,\sigma} \geq 1$.

We can also derive the upper bound of $\gamma_{\beta,\sigma}$,

$$\gamma_{\beta,\sigma} \leq \frac{\text{erf}(\frac{0.5/\sigma}{\sqrt{2}})}{\text{erf}(\frac{(0.5+\beta)/\sigma}{\sqrt{2}})}$$

$$\leq \frac{1}{1 - 2\beta}$$

where the first step comes from basic algebra, and the second step uses $\text{erf}(x) \approx \frac{2}{\sqrt{\pi}x}$.

$\square$

Here, we show our lemma on bounding the moment of privacy loss with our choice of truncated Gaussian noise.

**Lemma D.6** (Bound of $\alpha(\lambda)$ in sequential mechanism, formal version of Lemma 6.6). *Let $D, D' \in \mathcal{D}$ be $\eta$-close neighborhood polytope in Definition 1.2. Suppose that $f : \mathcal{D} \to \mathbb{R}^n$ with $\|f(D) - f(D')\|_2 \leq \beta$. Let $z \in \mathbb{R}^n$ be a truncated Gaussian noise vector in Definition D.4. Let $\sigma = \min \sigma_i$ and $\sigma \geq \beta$.*

*Then for any positive integer $\lambda \leq 1/4\gamma_{\beta,\sigma}$, there exists $C_0 > 0$ such that the mechanism $\mathcal{M}(d) = f(d) + z$ satisfies*

$$\alpha_{\mathcal{M}}(\lambda) \leq \frac{C_0\lambda(\lambda + 1)\beta^2\gamma_{\beta,\sigma}^2}{\sigma^2} + O(\beta^3\lambda^3\gamma_{\beta,\sigma}^3/\sigma^3)$$

*Proof.* Since $D, D'$ are neighborhood dataset, we can fix $D'$ and let $D = D' \cup \{D_i\}_{i \in [n]}$. Without loss of generality, we can assume $f(D_n) = \beta$ and for any $i \in [n-1]$, $f(D_i) = 0$. Thus, $\mathcal{M}(D)$ and $\mathcal{M}(D')$ have identical distributions other than the last coordinate. Then, we reduce it to a problem of one dimension.

Let $\mu_0(z_n)$ denote the probability density function of $\mathcal{N}^T(0, \sigma^2, [-0.5, 0.5])$

And let $\mu_1(z_n)$ denote the probability density function of $\mathcal{N}^T(\beta, \sigma^2, [-0.5, 0.5])$

Thus we have,

$$\mathcal{M}(D') \sim \mu_0(z_n),$$
$$\mathcal{M}(D) \sim \mu_1(z_n)$$

Recall Definition 6.3,

$$\alpha_{\mathcal{M}}(\lambda; \mathsf{aux}, D, D') = \log \mathop{\mathbb{E}}_{o \sim \mathcal{M}(\mathsf{aux}, D)} [\exp \lambda c(o; \mathcal{M}, \mathsf{aux}, D, D')]$$

And recall Definition 6.2,

$$c(o; \mathcal{M}, D, D') = \log \frac{\Pr[\mathcal{M}(D) = o]}{\Pr[\mathcal{M}(D') = o]}$$

We omit aux in Definition 6.2 because $\mathcal{M}$ here does not involve any auxiliary input.

Substitute $\mu_0$ and $\mu_1$ into $c(o; \mathcal{M}, D, D')$, we get

$$c(o; \mu_0, \mu_1, D, D') = \log \frac{\Pr[\mu_1 = o]}{\Pr[\mu_0 = o]} \tag{3}$$

Plug Eq. (3) into Definition 6.3, we get

$$\alpha_{\mathcal{M}}(\lambda; D, D') = \log \mathop{\mathbb{E}}_{o \sim \mathcal{M}(d)} [\exp(\lambda \log \frac{\Pr[\mu_1 = o]}{\Pr[\mu_0 = o]})] \tag{4}$$

Thus,

$$\alpha_{\mathcal{M}}(\lambda; D, D') = \log \mathop{\mathbb{E}}_{z_n \sim \mu_1} [(\mu_1(z_n)/\mu_0(z_n))^\lambda]$$
$$\leq \mathop{\mathbb{E}}_{z_n \sim \mu_1} [(\mu_1(z_n)/\mu_0(z_n))^\lambda]$$

where the first step follows from simplifying Eq. (4), the second step uses the property of logarithm.

We want to show that

$$\mathop{\mathbb{E}}_{z_n \sim \mu_1} [(\mu_1(z_n)/\mu_0(z_n))^\lambda] \leq \alpha_{\mathcal{M}}(\lambda)$$
$$\text{and} \mathop{\mathbb{E}}_{z_n \sim \mu_0} [(\mu_0(z_n)/\mu_1(z_n))^\lambda] \leq \alpha_{\mathcal{M}}(\lambda)$$

for some explicit $\alpha_{\mathcal{M}}(\lambda)$ to be determined later.

Since in our setting, both $\mu_0(z_n)$ and $\mu_1(z_n)$ are Gaussian variables, we only need to bound one of them by symmetry of Gaussian.

We consider

$$\mathop{\mathbb{E}}_{z_n \sim \mu_1} [(\mu_1(z_n)/\mu_0(z_n))^\lambda] = \mathop{\mathbb{E}}_{z_n \sim \mu_0} [(\mu_1(z_n)/\mu_0(z_n))^{\lambda+1}].$$

The above equality is obtained by the change of variable method in probability theory.

Using binomial expansion, we have

$$\mathop{\mathbb{E}}_{z_n \sim \mu_0} [\mu_1(z_n)/\mu_0(z_n))^{\lambda+1}] = \mathop{\mathbb{E}}_{z_n \sim \mu_0} [(1 + (\mu_1(z_n) - \mu_0(z_n))/\mu_0(z_n))^{\lambda+1}]$$
$$= \sum_{t=0}^{\lambda+1} \binom{\lambda+1}{t} \mathop{\mathbb{E}}_{z_n \sim \mu_0} [(\frac{\mu_1(z_n) - \mu_0(z_n)}{\mu_0(z_n)})^t]. \tag{5}$$

where the first step utilizes basic algebra, and the second step is by binomial expansion.

The first term in Eq. (5) is 1 by simple algebra, and the second term is

$$\mathop{\mathbb{E}}_{z_n \sim \mu_0} [\frac{\mu_1(z_n) - \mu_0(z_n)}{\mu_0(z_n)}] = \int_{-0.5}^{0.5} \mu_0(z_n) \frac{\mu_1(z_n) - \mu_0(z_n)}{\mu_0(z_n)} \, dz_n$$

$$= \int_{-0.5}^{0.5} \mu_1(z_n) \, dz_n - \int_{-0.5}^{0.5} \mu_0(z_n) \, dz_n$$

$$= 1 - 1 = 0.$$

where the first step is from the definition of expectation, the second step stems from basic algebra, and the last step utilizes the property of probability density function.

Recall Lemma D.1, for any $a \in \mathbb{R}$, $\mathbb{E}_{z \sim \mu_0} \exp(2az/2\sigma^2) = \exp(a^2/2\sigma^2)$, thus

$$\mathop{\mathbb{E}}_{z_n \sim \mu_0} [(\frac{\mu_1(z_n) - \mu_0(z_n)}{\mu_0(z_n)})^2] = \mathop{\mathbb{E}}_{z_n \sim \mu_0} [(1 - \gamma_{\beta,\sigma} \cdot \exp(\frac{2z_n\beta}{2\sigma^2} - \frac{\beta^2}{2\sigma^2}))^2]$$

$$= 1 - 2\gamma_{\beta,\sigma} \mathop{\mathbb{E}}_{z_n \sim \mu_0} [\exp(\frac{2z_n\beta}{2\sigma^2} - \frac{\beta^2}{2\sigma^2})] + \mathop{\mathbb{E}}_{z_n \sim \mu_0} [\gamma_{\beta,\sigma}^2 \exp(\frac{4z_n\beta}{2\sigma^2} - \frac{2\beta^2}{2\sigma^2})]$$

$$= 1 - 2\gamma_{\beta,\sigma} \exp(\frac{\beta^2}{2\sigma^2}) \cdot \exp(\frac{-\beta^2}{2\sigma^2})) + \gamma_{\beta,\sigma}^2 \exp(\frac{4\beta^2}{2\sigma^2}) \cdot \exp(\frac{-2\beta^2}{2\sigma^2})$$

$$= \gamma_{\beta,\sigma}^2 \exp(\frac{\beta^2}{\sigma^2}) + 1 - 2\gamma_{\beta,\sigma} \tag{6}$$

where the first step comes from substituting the density function of $\mu_0$ and $\mu_1$, the second step follows from expanding the square, the third step comes from Lemma D.1, and the final step comes from basic algebra.

Thus, the third term in the binomial expansion Eq. (5)

$$\binom{1+\lambda}{2} \mathop{\mathbb{E}}_{z_n \sim \mu_0} [(\frac{\mu_1(z_n) - \mu_0(z_n)}{\mu_0(z_n)})^2] \le \frac{\lambda(\lambda+1)}{2} \mathop{\mathbb{E}}_{z_n \sim \mu_0} [(\frac{\mu_1(z_n) - \mu_0(z_n)}{\mu_0(z_n)})^2]$$

$$= \frac{\lambda(\lambda+1)}{2} (\gamma_{\beta,\sigma}^2 \exp(\frac{\beta^2}{\sigma^2}) + 1 - 2\gamma_{\beta,\sigma})$$

$$= \frac{\lambda(\lambda+1)}{2} (\gamma_{\beta,\sigma}^2 \exp(\frac{\beta^2}{\sigma^2}) + (\gamma_{\beta,\sigma} - 1)^2 - \gamma_{\beta,\sigma}^2)$$

$$\le \frac{\lambda(\lambda+1)}{2} (\gamma_{\beta,\sigma}^2 (\exp(\frac{\beta^2}{\sigma^2}) - 1) + (\gamma_{\beta,\sigma} - 1)^2)$$

$$\le \frac{\lambda(\lambda+1)\beta^2 \gamma_{\beta,\sigma}^2}{\sigma^2} + \frac{\lambda(\lambda+1)(\gamma_{\beta,\sigma} - 1)^2}{2}$$

where the first step utilizes the definition of combination, the second step is the result of Eq. (6), the third step is by basic algebra, the fourth step comes from combining like terms, and the final step follows from basic algebra.

By our choice of $\sigma \ge \beta$, there exists $C_0 > 0$ to bound the third term in Eq. (5)

$$\binom{1+\lambda}{2} \mathop{\mathbb{E}}_{z_n \sim \mu_0} [(\frac{\mu_1(z_n) - \mu_0(z_n)}{\mu_0(z_n)})^2] \le \frac{C_0\lambda(\lambda+1)\beta^2\gamma_{\beta,\sigma}^2}{\sigma^2}$$

To bound the remaining terms, we first note that by standard calculus, we can bound $|\mu_0(z_n) - \mu_1(z_n)|$ by separating it into 3 parts.

Firstly, we have the following bound for all $z_n \le 0$.

$$|\mu_0(z_n) - \mu_1(z_n)| = \frac{1}{\sigma} |\frac{\phi(z_n/\sigma)}{C_\sigma} - \frac{\phi((z_n - \beta)/\sigma)}{C_{\beta,\sigma}}|$$

$$= \frac{1}{\sigma} |\frac{1}{C_\sigma} \cdot \frac{1}{\sqrt{2\pi}\sigma} \exp(-\frac{z_n^2}{2\sigma^2}) - \frac{1}{C_{\beta,\sigma}} \cdot \frac{1}{\sqrt{2\pi}\sigma} \exp(-\frac{(z - \beta)^2}{2\sigma^2})|$$

$$= \frac{1}{\sqrt{2\pi}\sigma^2} |\frac{\exp(-z_n^2/2\sigma^2)}{C_\sigma} - \frac{\exp(-(z_n - \beta)^2/2\sigma^2)}{C_{\beta,\sigma}}|$$

$$= \frac{1}{\sqrt{2\pi\sigma^2}} \left| \frac{\exp(-z^2/2\sigma^2)}{C_\sigma} - \frac{\exp(-(z^2 - 2\beta z + \beta^2)/2\sigma^2)}{C_{\beta,\sigma}} \right|$$

$$= \frac{1}{\sqrt{2\pi\sigma^2}} \exp(-\frac{z_n^2}{2\sigma^2}) \left| \frac{1}{C_\sigma} - \frac{1}{C_{\beta,\sigma}} \cdot \exp(\frac{2\beta z_n - \beta^2}{2\sigma}) \right|$$

$$= \frac{1}{C_\sigma} \frac{1}{\sqrt{2\pi\sigma^2}} \exp(-\frac{z_n^2}{2\sigma^2}) \left| 1 - \frac{C_\sigma}{C_{\beta,\sigma}} \cdot \exp(\frac{2\beta z_n - \beta^2}{2\sigma}) \right|$$

$$= \mu_0(z_n) \left| 1 - \frac{C_\sigma}{C_{\beta,\sigma}} \cdot \exp(\frac{2\beta z_n - \beta^2}{2\sigma}) \right|$$

$$\leq \mu_0(z_n) \frac{C_\sigma}{C_{\beta,\sigma}} \cdot \frac{-2\beta z_n + \beta^2}{2\sigma^2}$$

$$\leq \frac{-C_\sigma(\beta z_n - \beta^2)\mu_0(z_n)}{C_{\beta,\sigma}\sigma^2} \tag{7}$$

where the first step comes from the definition of $\mu_0, \mu_1$, the second step is from the probability density function of truncated Gaussian, the third step stems from basic algebra, the fourth step is from expanding the square, the fifth step uses basic algebra, the sixth step is from basic algebra, the seventh step derives from the definition of $\mu_0$, the eighth step follows by Taylor series, and the final step derives from basic algebra.

Similarly, for all $z_n$ such that $\beta \leq z_n \leq 0.5$, we have

$$|\mu_0(z_n) - \mu_1(z_n)| = \frac{1}{\sqrt{2\pi\sigma^2}} \left( \frac{1}{C_{\beta,\sigma}} \exp(-\frac{(z_n - \beta)^2}{2\sigma^2}) - \frac{1}{C_\sigma} \exp(-\frac{z_n^2}{2\sigma^2}) \right)$$

$$= \frac{1}{\sqrt{2\pi\sigma^2}} \left( \frac{1}{C_{\beta,\sigma}} \exp(-\frac{(z_n - \beta)^2}{2\sigma^2}) - \frac{1}{C_\sigma} \exp(-\frac{(z_n - \beta)^2 + 2z_n\beta - \beta^2}{2\sigma^2}) \right)$$

$$= \frac{1}{C_{\beta,\sigma}} \frac{1}{\sqrt{2\pi\sigma^2}} \exp(\frac{-(z_n - \beta)^2}{2\sigma^2})(1 - \frac{C_{\beta,\sigma}}{C_\sigma} \exp(-\frac{2z_n\beta - \beta^2}{2\sigma^2}))$$

$$= \mu_1(z_n)(1 - \frac{C_{\beta,\sigma}}{C_\sigma} \exp(-\frac{2z_n\beta - \beta^2}{2\sigma^2}))$$

$$\leq \mu_1(z_n) \frac{2z_n\beta - \beta^2}{2\sigma^2} \frac{C_{\beta,\sigma}}{C_\sigma}$$

$$\leq \frac{\beta(z_n - \beta)\mu_1(z_n)}{\sigma^2} \frac{C_{\beta,\sigma}}{C_\sigma}$$

$$\leq \frac{C_{\beta,\sigma} z_n \beta \mu_1(z_n)}{C_\sigma \sigma^2} \tag{8}$$

where the first step derives from the definition of $\mu_0, \mu_1$, the second step is from the probability density function of truncated Gaussian, the third step uses basic algebra, the fourth step comes from the definition of $\mu_1$, the fifth step is the result of Taylor series, the sixth step follows by basic algebra, and the final step comes from reorganization.

For all $z_n$ such that $0 \leq z_n \leq \beta$, we have

$$|\mu_0(z_n) - \mu_1(z_n)| \leq \frac{C_\sigma \beta^2 \mu_0(z_n)}{C_{\beta,\sigma}\sigma^2} \tag{9}$$

where we derive the bound by pluging $z_n = 0$ in Eq. (7).

We can then divide the expectation into three parts and bound them individually,

$$\mathbb{E}_{z_n \sim \mu_0} \left[ \left( \frac{\mu_1(z_n) - \mu_0(z_n)}{\mu_0(z_n)} \right)^t \right] \leq \int_{-0.5}^{0} \mu_0(z_n) \left| \left( \frac{\mu_1(z_n) - \mu_0(z_n)}{\mu_0(z_n)} \right)^t \right| dz_n$$

$$+ \int_{0}^{\beta} \mu_0(z_n) \left| \left( \frac{\mu_1(z_n) - \mu_0(z_n)}{\mu_0(z_n)} \right)^t \right| dz_n$$

$$+ \int_{\beta}^{0.5} \mu_0(z_n) |(\frac{\mu_1(z_n) - \mu_0(z_n)}{\mu_0(z_n)})^t| dz.$$

Notice the fact that $\mathbb{E}_{z_n \sim \mathcal{N}(0,\sigma^2)}[|z_n|^t] \leq \sigma^t(t-1)!!$ by Gaussian moments. Therefore, the first term can then be bounded by

$$\int_{-0.5}^{0} \mu_0(z_n) |(\frac{\mu_1(z_n) - \mu_0(z_n)}{\mu_0(z_n)})^t| dz_n \leq \int_{-0.5}^{0} \mu_0(z_n) |(\frac{-C_\sigma(\beta z_n - \beta^2)}{C_{\beta,\sigma}\sigma^2})^t| dz$$

$$\leq \frac{C_\sigma^t \beta^t}{C_{\beta,\sigma}^t \sigma^{2t}} \int_{-0.5}^{0} \mu_0(z_n) |(z_n - \beta)^t| dz_n$$

$$= \frac{C_\sigma^{t-1} \beta^t}{C_{\beta,\sigma}^t \sigma^{2t}} \int_{-0.5}^{0} \phi(z_n/\sigma) |(z_n - \beta)^t| dz$$

$$\leq \frac{(2\beta)^t C_\sigma^{t-1}(t-1)!!}{2 C_{\beta,\sigma}^t \sigma^t}$$

where the first step is the result of Eq. (7), the second step is from factoring out constants, the third step follows by the definition of $\mu_0$ and $C_\sigma$, and the final step comes from Gaussian moments bound.

The second term is at most

$$\int_{0}^{\beta} \mu_0(z_n) |(\frac{\mu_1(z_n) - \mu_0(z_n)}{\mu_0(z_n)})^t| dz_n \leq \int_{0}^{\beta} \mu_0(z_n) |(\frac{C_\sigma \beta^2}{C_{\beta,\sigma}\sigma^2})^t| dz_n$$

$$= \frac{C_\sigma^t \beta^{2t}}{C_{\beta,\sigma}^t \sigma^{2t}} \int_{0}^{\beta} \mu_0(z_n) dz_n$$

$$= \frac{C_\sigma^{t-1} \beta^{2t}}{C_{\beta,\sigma}^t \sigma^{2t}} \int_{0}^{\beta} \phi(z_n/\sigma) dz_n$$

$$\leq \frac{C_\sigma^{t-1} \beta^{2t}}{C_{\beta,\sigma}^t \sigma^{2t}}$$

where the first step derives from Eq. (9), the second step is from factoring out the constants, the third step is from the definition of $\mu_0$ and $C_\sigma$, and the final step stems from the property of probability density function.

Similarly, the third term is at most

$$\int_{\beta}^{0.5} \mu_0(z_n) |(\frac{\mu_1(z_n) - \mu_0(z_n)}{\mu_0(z_n)})^t| dz_n \leq \int_{\beta}^{0.5} \mu_0(z_n) |(\frac{C_{\beta,\sigma} z_n \beta \mu_1(z_n)}{C_\sigma \sigma^2 \mu_0(z_n)})^t| dz_n$$

$$= \frac{C_{\beta,\sigma}^t \beta^t}{C_\sigma^t \sigma^{2t}} \int_{\beta}^{0.5} \mu_0(z_n) |(\frac{z_n \mu_1(z_n)}{\mu_0(z_n)})^t| dz_n$$

$$= \frac{C_{\beta,\sigma}^t \beta^t}{C_\sigma^t \sigma^{2t}} \int_{\beta}^{0.5} (\frac{C_\sigma}{C_{\beta,\sigma}})^t \cdot \mu_0(z_n) \exp(\frac{2\beta t z_n - \beta^2 t}{2\sigma^2}) z_n^t dz_n$$

$$= \frac{\beta^t}{\sigma^{2t}} \int_{\beta}^{0.5} \mu_0(z_n) \exp(\frac{2\beta t z_n - \beta^2 t}{2\sigma^2}) z_n^t dz_n$$

$$= \frac{\beta^t}{C_\sigma \sigma^{2t}} \int_{\beta}^{0.5} \phi(z/\sigma) \exp(\frac{2\beta t z_n - \beta^2 t}{2\sigma^2}) z_n^t dz_n$$

$$\leq \frac{\beta^t \exp(\beta^2(t^2 - t)/2\sigma^2)}{C_\sigma \sigma^{2t}} \int_{0}^{0.5} \phi(\frac{z_n - \beta t}{\sigma}) z_n^t dz_n$$

$$\leq \frac{(2\beta)^t \exp(\beta^2(t^2 - t)/2\sigma^2)(\sigma^t(t-1)!! + (\beta t)^t)}{2 C_\sigma \sigma^{2t}}$$

where the first step is the result of Eq. (8), the second step derives from factoring out constants, the third step is from plugging the density function of $\mu_0$ and $\mu_1$ into the expression, the fourth

step is by simplification, the fifth step derives from the definition of $\mu_0$ the sixth step comes from $U$-substitution in calculus, and the final step follows by Gaussian moment bound.

Finally, we can show that our choice of parameters is valid. By plugging above bounds into Eq. (5), we observe that with constraints on $\sigma, \beta, \lambda, \gamma_{\beta,\sigma}$ in the Lemma statement, it's obvious that higher-order terms with $t > 3$ will be dominated by the $t = 3$ term. Therefore, we conclude that

$$\alpha_{\mathcal{M}}(\lambda) \leq \frac{C_0 \lambda (\lambda + 1) \beta^2 \gamma_{\beta,\sigma}^2}{\sigma^2} + O(\beta^3 \lambda^3 \gamma_{\beta,\sigma}^3 / \sigma^3)$$

$\square$

## D.3. Privacy of Fast John Ellipsoid Algorithm with Truncated Noise

By combining the above moment bounds in Lemma D.6 and tail bounds in Theorem D.8, we can proceed to derive our main theorem, which demonstrates that the John Ellipsoid algorithm is differentially private.

**Theorem D.7** (John Ellipsoid DP Main Theorem, formal version of Theorem 6.7). *Suppose the input polytope in Algorithm 1 represented by $A \in \mathbb{R}^{n \times d}$ satisfies $\sigma_{\max}(A) \leq \mathrm{poly}(n)$ and $\sigma_{\min}(A) \geq 1/\mathrm{poly}(n)$. Under our definition of $\epsilon_0$-close neighborhood polytope, described in Definition 1.2, let $L$ be the Lipschitz of such polytope. Then there exists constants $c_1$ and $c_2$ so that given number of iterations $T$, for any $\epsilon \leq c_1 T L^2 \epsilon_0^2 (1 - 2L\epsilon_0)^{-1}$, Algorithm 1 is $(\epsilon, \delta)$-differentially private for any $\delta > 0$ if we choose*

$$\sigma \geq c_2 \frac{L\epsilon_0 \sqrt{T \log(1/\delta)}}{(1 - 2L\epsilon_0)\epsilon}$$

*Proof.* According to Lemma D.6, we have $\sigma \geq \beta$ and $\lambda \leq 1/4\gamma_{\beta,\sigma}$. According to Theorem C.15, we can set $\beta = L\epsilon_0$.

According to Lemma D.5 and substituting $\beta$ with $L\epsilon_0$, we have

$$\gamma_{L\epsilon_0,\sigma} \leq \frac{1}{1 - 2L\epsilon_0}$$

By the composability of moment bounds in Theorem D.8 and Lemma D.6, we have the following by substituting $\gamma_{\beta,\sigma}$ with $\gamma_{L\epsilon_0,\sigma}$ and $\beta$ with $L\epsilon_0$

$$\alpha(\lambda) \leq T C_0 L^2 \epsilon_0^2 \lambda^2 \gamma_{L\epsilon_0,\sigma}^2 \sigma^{-2}$$

According to Theorem D.8, we need to ensure the followings so that Algorithm 1 is $(\epsilon, \delta)$-differentially private.

$$T C_0 L^2 \epsilon_0^2 \lambda^2 \gamma_{L\epsilon_0,\sigma}^2 \sigma^{-2} \leq \lambda\epsilon/2,$$
$$\exp(-\lambda\epsilon/2) \leq \delta.$$

Therefore, when $\epsilon = c_1 T L^2 \epsilon_0^2 (1 - 2L\epsilon_0)^{-1}$, we can satisfy all of these conditions by setting

$$\sigma \geq c_2 \frac{L\epsilon_0 \sqrt{T \log(1/\delta)}}{(1 - 2L\epsilon_0)\epsilon}$$

$\square$

## D.4. Composition Lemma for Adaptive Mechanisms

In this section, we list the powerful composition lemma for the adaptive mechanism proposed in [89], which we utilized to demonstrate the privacy guarantee on Algorithm 1.

**Theorem D.8** (Theorem 2 in [89]). *Let $k$ be an integer, representing the number of sequential mechanisms in $\mathcal{M}$. We define $\alpha_{\mathcal{M}}(\lambda)$ as*

$$\alpha_{\mathcal{M}}(\lambda) := \max_{\mathsf{aux}, D, D'} \alpha_{\mathcal{M}}(\lambda; \mathsf{aux}, D, D'),$$

*where the maximum is taken over all auxiliary inputs and neighboring databases $D, D'$. Then*

1. **[Composability]** *Suppose that a mechanism $\mathcal{M}$ consists of a sequence of adaptive mechanisms $\mathcal{M}_1, \cdots, \mathcal{M}_k$ where $\mathcal{M}_i : \prod_{j=1}^{i-1} \mathcal{R}_j \times \mathcal{D} \rightarrow \mathcal{R}_i$. Then, for any $\lambda > 0$*

$$\alpha_{\mathcal{M}}(\lambda) \leq \sum_{i=1}^{k} \alpha_{\mathcal{M}_i}(\lambda)$$

2. **[Tail bound]** *For any $\epsilon > 0$, we have the mechanism $\mathcal{M}$ is $(\epsilon, \delta)$-differentially private where*

$$\delta = \min_{\lambda} \exp(\alpha_{\mathcal{M}}(\lambda) - \lambda\epsilon)$$

# E. Convergence Proof for DP John Ellipsoid Algorithm

Firstly, in Section E.1, we include the previous proposition and corollary used to show the convergence of our John Ellipsoid algorithm. Then, in Section E.2, we introduce our telescoping lemma. In Section E.3, we demonstrate the high probability bound in the error caused by leverage score sampling and sketching in Algorithm 1. Next, in Section E.4, we discuss the high probability bound on the error caused by adding truncated Gaussian noise. Then, we show the upper bound of $\phi$ in Section E.5. Finally, we demonstrate the convergence and correctness of Algorithm 1 in Section E.6.

## E.1. Previous Work in John Ellipsoid Algorithm

In this subsection, we list some findings in previous work that help us to show the convergence of Algorithm 1.

**Proposition E.1** (Bound on $\widehat{w}^{(k)}$, Proposition C.1 in [10]). *For completeness we define $\widehat{w}^{(1)} = w^{(1)}$. For $k \in [T]$ and $i \in [n], 0 \leq \widehat{w}_i^{(k)} \leq 1$. Moreover, $\sum_{i=1}^{n} \widehat{w}_i^{(k)} = d$.*

**Corollary E.2** (Corollary 8.5 in [9]). *Let $\xi_0$ denote the accuracy parameter defined as Algorithm 1. Let $\delta_0$ denote the failure probability.*

*Then we have with probability $1 - \delta_0$, the inequality below holds for all $i \in [n]$*

$$(1 - \xi)\widetilde{w}_i \leq \widehat{w}_i \leq (1 + \xi)\widetilde{w}_i.$$

## E.2. Telescoping Lemma

In this subsection, we demonstrate the telescoping lemma, a technique we choose to show the convergence proof. We demonstrate the convergence and accuracy of Algorithm 1 by deriving the upper bound of the logarithm of the leverage score.

Now, we present the telescoping lemma for our fast JE algorithm. While the telescoping lemma (Lemma 6.1 [9]) deals with sketching and leverage score sampling, our lemma considers the circumstance where the truncated Gaussian noise is included to ensure the privacy of John Ellipsoid algorithm.

**Lemma E.3** (Telescoping, Algorithm 1, formal version of Lemma 7.1). *Let $T$ denote the number of iterations in the main loop in our fast JE algorithm. Let $u$ be the vector obtained in Algorithm 1. Thus for each $i \in [n]$, we have*

$$\phi_i(u) \leq \frac{1}{T}\log\frac{n}{d} + \frac{1}{T}\sum_{k=1}^{T}\log\frac{\widehat{w}_{k,i}}{\widetilde{w}_{k,i}} + \frac{1}{T}\sum_{k=1}^{T}\log\frac{\widetilde{w}_{k,i}}{\overline{w}_{k,i}} + \frac{1}{T}\sum_{k=1}^{T}\log\frac{\overline{w}_{k,i}}{w_{k,i}}$$

*Proof.* We define $u$ and $w$ as the following

$$u := (u_1, u_2, \cdots, u_n).$$

For $k \in [T-1]$, we define

$$w_k := (w_{k,1}, \cdots, w_{k,n})$$

and

$$w_{k+1} := (w_{k,1}h_1(w_k), \cdots, w_{k,n}h_n(w_k)).$$

Now we consider $\phi_i(u)$, defined in Lemma 3.9

$$
\begin{aligned}
\phi_i(u) &= \phi_i(\frac{1}{T}\sum_{k=1}^{T} w_k) \\
&\leq \frac{1}{T}\sum_{k=1}^{T} \phi_i(w_k) \\
&= \frac{1}{T}\sum_{k=1}^{T} \log h_i(w_k) \\
&= \frac{1}{T}\sum_{k=1}^{T} \log \frac{\widehat{w}_{k+1,i}}{w_{k,i}} \\
&= \frac{1}{T}\sum_{k=1}^{T} \log \frac{\widehat{w}_{k+1,i}}{\widehat{w}_{k,i}} \cdot \frac{\widehat{w}_{k,i}}{\widetilde{w}_{k,i}} \cdot \frac{\widetilde{w}_{k,i}}{\overline{w}_{k,i}} \cdot \frac{\overline{w}_{k,i}}{w_{k,i}} \\
&= \frac{1}{T}(\sum_{k=1}^{T} \log \frac{\widehat{w}_{k+1,i}}{\widehat{w}_{k,i}} + \sum_{k=1}^{T} \log \frac{\widehat{w}_{k,i}}{\widetilde{w}_{k,i}} + \sum_{k=1}^{T} \log \frac{\widetilde{w}_{k,i}}{\overline{w}_{k,i}} + \sum_{k=1}^{T} \log \frac{\overline{w}_{k,i}}{w_{k,i}})
\end{aligned}
$$

where the first step is from the definition of $u$, the second step is from Lemma 3.9, the third step utilizes the definition of $h_i$, the fourth step derives from the definition of $w_{k+1}$, the fifth step derives from basic algebra, the last step is from logarithm arithmetic.

$\square$

## E.3. High Probability Bound of Sketching and Leverage Score Sampling

In this subsection, we first demonstrate the high probability bound for leverage score sampling. Then, we demonstrate the high probability bound for the error of sketching in Algorithm 1.

**Lemma E.4.** *Let $\delta$ be the failure probability. Then for any $\xi_0 \in [0, 0.1]$, if $T > \xi_0^{-1}\log(n/d)$, with probability $1 - \delta$, we have*

$$\frac{1}{T}\sum_{k=1}^{T} \log \frac{\widehat{w}_{k,i}}{\widetilde{w}_{k,i}} \leq \xi_0$$

*Proof.* By Corollary E.2, we have with probability of $1 - \delta$, we can derive

$$
\begin{aligned}
\frac{1}{T}\sum_{k=1}^{T} \log \frac{\widehat{w}_{k,i}}{\widetilde{w}_{k,i}} &\leq \log(1 + \xi_0) \\
&\leq \xi_0
\end{aligned}
$$

$\square$

Now, we proceed to derive the high probability bound of sketching. Here, we list a Lemma from [9] on the error of sketching.

**Lemma E.5** (Lemma 6.3 in [9])**.** *We have the following for failure probability for sketching $\delta \in [0, 0.1]$*

$$\Pr[\frac{\widetilde{w}_{k,i}}{\overline{w}_{k,i}} \geq 1 + \xi] \leq \frac{(\frac{n}{d})^{\frac{\alpha}{T}} e^{\frac{4\alpha}{s}}}{(1 + \xi)^{\alpha}}.$$

*Furthermore, with appropriate $s, T$, we have the following hold with large $n$ and $d$:*

$$\Pr[\frac{\widetilde{w}_{k,i}}{\overline{w}_{k,i}} \geq 1 + \xi] \leq \frac{\delta}{n}$$

Applying the above lemma, we derive the error caused by sketching in our setting.

**Lemma E.6.** *Let $\delta$ be the failure probability. Then for any $\xi_1 \in [0, 0.1]$, if $T > \xi_1^{-1} \log(n/d)$, we have the following hold for all $i \in [n]$ with probability $1 - \delta$:*

$$\frac{1}{T} \sum_{k=1}^{T} \log \frac{\widetilde{w}_{k,i}}{\overline{w}_{k,i}} \leq \xi_1$$

*Proof.* By Lemma E.5, we have with high probability

$$\log \frac{\widetilde{w}_{k,i}}{\overline{w}_{k,i}} \leq \log(1 + \xi_1).$$

By the central limit theorem, with probability $1 - \delta$, we have

$$\frac{1}{T} \sum_{k=1}^{T} \log \frac{\widetilde{w}_{k,i}}{\overline{w}_{k,i}} \leq \log(1 + \xi_1)$$

$$\leq \xi_1$$

$\square$

## E.4. High Probability Bound of Adding Truncated Gaussian Noise

In this subsection, we demonstrate the error bound of adding Truncated Gaussian noise. Here is the fact about the upper bound of the logarithm.

**Fact E.7.** *For $z \in [-0.5, 0.5]$, then*

$$\log(\frac{1}{1 + z}) \leq 2|z|.$$

*Proof.* It is clear that $\log(\frac{1}{1+z}) \leq 2|z|$ holds when $z \in [0, 0.5]$ since $\log(\frac{1}{1+z}) \leq 0$ and $2|z| \geq 0$ for $z \in [0, 0.5]$. Next, we show that this also holds when $z \in [-0.5, 0]$. Let $h(z) = -2z - \log(\frac{1}{1+z})$. Then $h'(z) = \frac{1}{1+z} - 2 \leq 0$. Hence $h$ is decreasing on $[-0.5, 0]$ and $h(-0.5) = 1 - \log(2) > 0$. Hence $\log(\frac{1}{1+z}) \leq 2|z|$ when $z \in [-0.5, 0]$. $\square$

Then, we derive the upper bound on the expectation of truncated Gaussian noise.

**Lemma E.8.** *For any $\sigma \in [0, 0.1]$, let $Z \sim \mathcal{N}^T(0, \sigma^2; [-0.5, 0.5])$. Then*

$$\mathbb{E}[|Z|] \leq \sigma.$$

*Proof.* The pdf of $Z$ is

$$f(z) = \frac{1}{\sigma} \frac{\phi(z/\sigma)}{\Phi(0.5/\sigma) - \Phi(-0.5/\sigma)} \text{ for } z \in [-0.5, 0.5]$$

where $\phi$ and $\Phi$ are pdf and cdf of the standard Gaussian.

Then we have

$$\mathbb{E}[|Z|] = \int_{-0.5}^{0.5} |z| \frac{1}{\sigma} \frac{\phi(z/\sigma)}{\Phi(0.5/\sigma) - \Phi(-0.5/\sigma)} \mathrm{d}z$$

$$= \int_0^{0.5} 2z \frac{1}{\sigma} \frac{\phi(z/\sigma)}{\Phi(0.5/\sigma) - \Phi(-0.5/\sigma)} \mathrm{d}z$$

$$= \frac{1}{\sigma} \frac{2}{\Phi(0.5/\sigma) - \Phi(-0.5/\sigma)} \int_0^{0.5} z\phi(z/\sigma) \mathrm{d}z$$

$$= \frac{2\sigma}{\Phi(0.5/\sigma) - \Phi(-0.5/\sigma)} \int_0^{0.5/\sigma} u\phi(u) \mathrm{d}u$$

where the first step follows from the definition of expectation, the second step follows from the symmetry of $Z$, the third step follows from simple rearranging, and the last step follows from $u :=$ $z/\sigma$.

Now, we evaluate the following integral:

$$\int_0^{0.5/\sigma} u\phi(u) \mathrm{d}u = \frac{1 - e^{-(0.5/\sigma)^2/2}}{\sqrt{2\pi}} \leq \frac{1}{\sqrt{2\pi}}.$$

From standard Gaussian table, for $\sigma \leq 0.1$, we have

$$\Phi(0.5/\sigma) - \Phi(-0.5/\sigma) \geq \Phi(5) - \Phi(-5) \geq 0.999.$$

Hence

$$\mathbb{E}[|Z|] \leq \frac{\sqrt{2/\pi}}{\Phi(0.5/\sigma) - \Phi(-0.5/\sigma)} \sigma \leq \sigma.$$

$\square$

Combining previous bounds, we can show that the error is caused by adding noise in one certain iteration.

**Lemma E.9.** *We have*

$$\mathbb{E}[\log \frac{\overline{w}_{k,i}}{w_{k,i}}] \leq 2\sigma$$

*where the randomness is derived from the truncated Gaussian noise in $w_{k,i}$.*

*Proof.* By Fact E.7 and Lemma E.8, we have

$$\mathbb{E}[\log \frac{\overline{w}_{k,i}}{w_{k,i}}] = \mathbb{E}[\log \frac{\overline{w}_{k,i}}{\overline{w}_{k,i}(1 + z_{k,i})}]$$

$$= \mathbb{E}[\log \frac{1}{1 + z_{k+1,i}}]$$

$$\leq \mathbb{E}[2 \cdot |z_{k+1,i}|]$$

$$\leq 2\sigma$$

where the first step is from $w_{k,i} := \overline{w}_{k,i}(1 + z_{k,i})$, the second step stems from basic algebra, the third step comes from Fact E.7, the last step follows from Lemma E.8. $\square$

Finally, applying concentration inequality, we can show the total error caused by truncated Gaussian noise in Algorithm 1.

**Lemma E.10.** *Let $\delta$ be the failure probability. Then for any $\xi_2 \in [0, 0.1]$, if $T > \xi_2^{-2} \log(1/\delta)$, with probability $1 - \delta$, we have*

$$\frac{1}{T} \sum_{k=1}^T \log \frac{\overline{w}_{k,i}}{w_{k,i}} \leq \xi_2.$$

*where the randomness is from the truncated Gaussian noise in $w_{k,i}$.*

*Proof.* The range of $\log \frac{\overline{w}_{k,i}}{w_{k,i}}$ is $[\log(2/3), \log 2]$. Applying Hoeffding's inequality shows that for $t > 0$,

$$\Pr[\sum_{k=1}^{T} \log \frac{\overline{w}_{k,i}}{w_{k,i}} \geq 2\sigma T + t] < \exp(-\frac{2t^2}{T(\log 3)^2}).$$

Pick $t = 2\sigma T$. Then we get

$$\Pr[\sum_{k=1}^{T} \log \frac{\overline{w}_{k,i}}{w_{k,i}} \geq 4\sigma T] < \exp(-\frac{8\sigma^2 T}{(\log 3)^2}).$$

It implies that if $T > \sigma^{-2} \log(1/\delta)$, with probability $1 - \delta$, we have

$$\sum_{k=1}^{T} \log \frac{\overline{w}_{k,i}}{w_{k,i}} \leq 4\sigma T.$$

Set $\xi_2 = 4\sigma$, we have

$$\frac{1}{T} \sum_{k=1}^{T} \log \frac{\overline{w}_{k,i}}{w_{k,i}} \leq \xi_2$$

$\square$

### E.5. Upper Bound of $\phi_i$

In this subsection, we derive the upper bound of $\phi_i$ by combining the error bound for truncated Gaussian noise and the error bound of leverage score sampling and sketching.

**Lemma E.11** ($\phi_i$)**.** *Consider the vector $u$ generated in Algorithm 1, and let the number of iterations in the main loop of the algorithm be $T$ and $s = 1000/\xi_1$. With probability $1 - \delta$, the following inequality holds for all $i \in [n]$*

$$\phi_i(u) \leq \frac{1}{T} \log(\frac{n}{d}) + \xi_0 + \xi_1 + \xi_2.$$

*Proof.* To begin with, by Lemma E.3, we have that

$$\phi_i(u) \leq \frac{1}{T} \log \frac{n}{d} + \frac{1}{T} \sum_{k=1}^{T} \log \frac{\widehat{w}_{k,i}}{\widetilde{w}_{k,i}} + \frac{1}{T} \sum_{k=1}^{T} \log \frac{\widetilde{w}_{k,i}}{\overline{w}_{k,i}} + \frac{1}{T} \sum_{k=1}^{T} \log \frac{\overline{w}_{k,i}}{w_{k,i}}$$

By Lemma E.4, we have with probability $1 - \delta/3$, for all $i \in [n]$:

$$\frac{1}{T} \sum_{k=1}^{T} \log \frac{\widehat{w}_{k,i}}{\widetilde{w}_{k,i}} \leq \xi_0$$

By Lemma E.6, with probability $1 - \delta/3$, the following holds for all $i \in [n]$:

$$\frac{1}{T} \sum_{k=1}^{T} \log \frac{\widetilde{w}_{k,i}}{\overline{w}_{k,i}} \leq \xi_1$$

Next, by Lemma E.10, with probability $1 - \delta/3$, for all $i \in [n]$:

$$\frac{1}{T} \sum_{k=1}^{T} \log \frac{\overline{w}_{k,i}}{w_{k,i}} \leq \xi_2.$$

Putting everything together, with $1 - \delta$, for all $i \in [n]$, we have

$$\phi_i(u) \leq \frac{1}{T} \log(\frac{n}{d}) + \xi_0 + \xi_1 + \xi_2.$$

$\square$

## E.6. Convergence Result

Finally, we proceed to the convergence main theorem, which demonstrates that Algorithm 1 could find a good approximation of John Ellipsoid with our choice of parameters.

**Theorem E.12** (Convergence main theorem, formal version of Theorem 7.2). *Let $u \in \mathbb{R}^n$ be the non-normalized output of Algorithm 1, $\delta_0$ be the failure probability. For all $\xi \in (0,1)$, when $T = \Theta(\xi^{-1} \log(n/d) + \xi^{-2} \log(1/\delta))$, we have*

$$\Pr[h_i(u) \leq (1 + \xi), \forall i \in [n]] \geq 1 - \delta_0$$

*In addition,*

$$\sum_{i=1}^{n} v_i = d$$

*Thus, Algorithm 1 finds $(1 + \xi)$-approximation to the exact John ellipsoid solution.*

*Proof.* We set

$$\xi_0 = \xi_1 = \xi_2 = \frac{\xi}{8}$$

and

$$T = \Theta(\xi^{-1} \log(n/\delta) + \sigma^{-2} \log(1/\delta))$$

By Lemma E.11, with succeed probability $1 - \delta$. We have for all $i \in [n]$,

$$
\begin{aligned}
\log h_i(u) &= \phi_i(u) \\
&\leq \frac{1}{T} \log(\frac{n}{d}) + \xi_0 + \xi_1 + \xi_2 \\
&\leq \frac{\xi}{2} \\
&\leq \log(1 + \xi)
\end{aligned}
$$

where the first step comes from the definition of $\phi$, the second step utilizes Lemma E.11, the third step comes from our choice of $T$ and $\xi$, and the last step follows by for all $\xi \in [0, 1]$, $\frac{\xi}{2} \leq \log(1 + \xi)$

Since we choose $v_i = \frac{d}{\sum_{j=1}^{n} u_j} u_i$, then we have

$$\sum_{i=1}^{n} v_i = d.$$

Next, we have

$$
\begin{aligned}
h_i(v) &= a_i^\top (A^\top V A)^{-1} a_i \\
&= a_i^\top (\frac{d}{\sum_{i=1}^{n} u_i} A^\top U A)^{-1} a_i \\
&= \frac{\sum_{i=1}^{n} u_i}{d} h_i(u) \\
&\leq (1 + \epsilon) \cdot h_i(u) \\
&\leq (1 + \epsilon)^2
\end{aligned}
$$

where the first step comes from the definition of $h_i(v)$, the second step follows by the definition of $V$, the third step comes from the definition of $h_i(u)$, the fourth step comes from Lemma E.1, the last step derives from $h_i(u) \leq 1 + \epsilon$.

Thus, we complete the proof. $\square$

# F. Proof of Main Theorem

In this section, we introduce our main theorem, which shows that while satisfying privacy guarantee, our Algorithm 1 achieves high accuracy and efficient running time.

**Theorem F.1** (Main Results, formal version of Theorem 4.1)**.** *Let $v \in \mathbb{R}^n$ be the result from Algorithm 1. Define L as in Theorem 5.1. For all $\xi, \delta_0 \in (0, 0.1)$, when $T = \Theta(\xi^{-2}(\log(n/\delta_0) + (L\epsilon_0)^{-2}))$, the inequality below holds for all $i \in [n]$:*

$$\Pr[h_i(v) \leq (1 + \xi)] \geq 1 - \delta_0$$

*In addition,*

$$\sum_{i=1}^{n} v_i = d$$

*Thus, Algorithm 1 gives $(1 + \xi)$-approximation to the exact John ellipsoid.*

*Furthermore, suppose the input polytope in Algorithm 1 represented by $A \in \mathbb{R}^{n \times d}$ satisfies $\sigma_{\max}(A) \leq \mathrm{poly}(n)$ and $\sigma_{\min}(A) \geq 1/\mathrm{poly}(n)$. Let $\epsilon_0 \leq O(1/\mathrm{poly}(n))$ be the closeness of the neighboring polytopes defined in Definition 1.2 and $L \leq O(\mathrm{poly}(n))$ be the Lipschitz defined in Theorem 5.1. Then there exists constant $c_1$ and $c_2$ so that given number of iterations $T$, for any $\epsilon \leq c_1 T L^2 \epsilon_0^2 (1 - 2L\epsilon_0)^{-1}$, Algorithm 1 is $(\epsilon, \delta)$-differentially private for any $\delta > 0$ if we choose*

$$\sigma \geq \frac{c_2 L \epsilon_0 \sqrt{T \log(1/\delta)}}{(1 - 2L\epsilon_0)\epsilon}$$

*The runtime of Algorithm 1 achieves $O((\mathrm{nnz}(A) + d^\omega)T)$, where $\omega \approx 2.37$ represent the matrix-multiplication exponent.*

*Proof.* It derives from Theorem E.12 and Theorem D.7. The running time analysis is the same as Algorithm 1 in [9]. □

