# OpenReview forum: "Fast John Ellipsoid Computation with Differential Privacy Optimization"
_CPAL.cc/2025/Proceedings_Track — CPAL 2025 (Proceedings Track) Oral_

### Official Review · Reviewer_1Zu8 · 2025-01-05
**Interesting idea, presentation and experimental evidence could be improved**

**Rating:** 7
**Confidence:** 4

**Review:**

The submission sets forth a differentially-private algorithm for the estimation of the John ellipsoid. The method combines ideas of leverage score sampling and truncated Gaussian noise to provide convergence and privacy guarantees.

Strenghts:
- The problem is well-motivated and of interest;
- The idea of including differential privacy is novel and of practical relevance.

Weaknesses:
- Presentation is hard to follow at times;
- Experimental evidence in support of the proposed algorithm would strengthen the claims made in the manuscript.

I will expand on a few comments below and I am looking forward to discussing with the authors!

---

1. Presentation. In its current form, presentation is hard to follow and the paper reads as a sequence of definitions and technical results with little discussion of their implications and intuitive understanding. In particular, certain paragraphs are written in the past tense, e.g. on Line 49 "In our work, we defined ...", but the definition has not been presented yet. Similarly, on Line 202, "we demonstrated that Algorithm 1 solved", but these results have not been presented at this point. This happens in several places throughout the manuscript.

It would be valuable to include more high-level discussion of the techniques used in the submission, in order for a first-time reader to follow the story better. For example, to a reader unfamiliar with DP literature, it may not be clear why using truncated Gaussian noise should lead to a privacy guarantee. Providing clear intuition behind these ideas is important to make the submission accessible to a broader audience who may want to use the proposed algorithm.

2. Lines 188-191 and Lines 196-197: At this point, it is unclear how using leverage score sampling and matrix sketching provides a speedup of the algorithm. Later in the paper, Lemma 7.1 states the complexity of the algorithm. Could the authors expand on this claim, providing evidence that the proposed algorithm beats existing alternatives in terms of runtime and/or computational complexity? What level of speedup is achieved by the proposed method?

3. Definition 6.2. It is unclear what the output of the mechanism is. Its range is defined as $\mathcal{R}$. What objects are being considered in this submission? Is $o \in \mathcal{R}$ a predictor? What information is contained in the auxiliary input aux? What is it in practice, and why is it needed for the contributions of the submission? What is the probability in Definition 6.2 taken over? The dataset, or the output of the mechanism, or both? Could the authors clarify these statements?

4. Experimental results are not present. The theoretical analysis presented in the submission is valuable and important, but it needs to be substantiated with experimental validation and comparison with existing methods.

---

**Minor comments**
- Line 23: Typo in "Monte Carlo"
- Definition 1.1: Should the polytope be defined with the row vectors of $A$ instead of the columns? Otherwise the inner product would be taken between a vector with $n$ entries and one with $d$ entries.
- Line 37: "for each round of sensitive pay-off value" is unclear. Could the authors expand on this sentence?
- Line 38: Typo in "the the"
- Lines 46-50: this paragraph feels out of place and it is unclear. Why does the general definition of neighborhood dataset fail to work? How can the definition of $\epsilon_0$-close ensure privacy of the Algorithm?
- Line 114: repetition in "by leveraging sketching and leverage score sampling"
- Line 133: typo in "we denote their [inner product] as"
- Lines 133-140: Dimensions change several times throughout this paragraph, making it hard to follow indices and what they refer to. For example, $A$ is in $n \times d$, but $x$ is in $k \times 1$, so dimensions in $Ax$ do not match. Also, $x,y$ are defined as vectors in $d$, $n$, and $k$ dimensions.
- Line 148: typo in "Ellipsoid solution have"
- Line 153: is $A$ a different matrix from the design matrix $A$? If so, what matrix is it?
- Line 210: "let $\sigma$ be the noise scale". What noise scale is being referred here? Also, $\sigma$ is already used to indicate the singular values of the matrix $A$.
- Lines 217-221: referencing particular lines within the proposed algorithm adds little to presentation of the results.
- Line 235: "about the sequential mechanism" is unclear. What sequential mechanism is being referred to here?
- Line 240: similar to previous point about "for the sequential mechanism"

---

### Official Review · Reviewer_WJNo · 2025-01-11
**Simple and clever approach to compute John Ellipsoid while maintaining privacy**

**Rating:** 8
**Confidence:** 3

**Review:**

This paper provides a new algorithm that computes the John Ellipsoid. This algorithm converges to a (1 + ξ)-approximation of the optimal John Ellipsoid while providing differential privacy.
Comparing to previous methods of John Ellipsoid computation, this new algorithm utilizes the so called sketching matrix to reduce the dimension of the matrix representation of the polytope and adds random truncated Gaussian noise to ensure privacy.
There are other fast computation methods of the John Ellipsoid but this paper is the first to consider the privacy issue and the approach is very simple.
The structure of the paper is nice with solid proofs. I would love to see some simulation/application results though.

---

### Official Review · Reviewer_zqQH · 2025-01-13
**Paper Review**

**Rating:** 7
**Confidence:** 3

**Review:**

**Summary**

This paper provides the first differentially private algorithm for fast John ellipsoid computation. The paper additionally provides a thorough theoretical analysis of the algorithm's convergence and privacy properties.

**Quality/Clarity**
The paper is very well written and easy to read. The problem this paper aims to answer is clearly defined along with the contributions.

**Strengths**
- The paper does an excellent job presenting all the background information needed to understand the problem and results.
- The paper provides an algorithm and then demonstrates how it 1) satisfies DP, 2) has high accuracy and 3) is efficient. I appreciate the thorough analysis of the algorithm.

**Weaknesses**
- While the paper provides strong theoretical results there are no experiments. Could the authors perform some experiments demonstrating that their algorithm exhibits all the desirable properties the paper states it does.

**Minor comments**
- Line 223: "We stated the following theorem" should be "We state the following theorem"

---

### Meta-Review · Area_Chair_jSSL · 2025-02-02

**Recommendation:** Accept (Poster)
**Confidence:** 5

**Metareview:**

This paper proposes a differentially-private algorithm for the estimation of the John ellipsoid.
It is a purely theoretical paper with no experiments showcasing the proposed algorithm.
The reviewers valued the novelty of the paper and the theoretical contributions.

---

### Decision · Program_Chairs · 2025-02-11

Accept (Oral)